# Highly branched isoprenoids for Southern Ocean sea ice reconstructions: a pilot study from the Western Antarctic Peninsula

Maria-Elena Vorrath[1], Juliane Müller[1,2,3], Oliver Esper[1], Gesine Mollenhauer[1,2], Christian Haas[1], Enno Schefuß[2], Kirsten Fahl[1]

[1]Alfred Wegener Institute, Helmholtz Centre for Polar and Marine Research, Bremerhaven, Germany

[2]MARUM – Center for Marine Environmental Sciences, University of Bremen, Germany

[3]Department of Geosciences, University of Bremen, Germany

*Correspondence to*: Maria-Elena Vorrath, maria-elena.vorrath@awi.de

**Abstract**. Organic geochemical and micropaleontological analyses of surface sediments collected in the southern Drake Passage and the Bransfield Strait, Antarctic Peninsula, enable a proxy-based reconstruction of recent sea ice conditions in this climate sensitive area. We study the distribution of the sea ice biomarker $IPSO_{25}$, and biomarkers of open marine environments such as more unsaturated highly branched isoprenoid alkenes and phytosterols. Comparison of the sedimentary distribution of these biomarker lipids with sea ice data obtained from satellite observations and diatom-based sea ice estimates provide for an evaluation of the suitability of these biomarkers to reflect recent sea surface conditions. The distribution of $IPSO_{25}$ supports earlier suggestions that the source diatom seems to be common in near-coastal environments characterized by an annually recurring sea ice cover, while the distribution of the other biomarkers is highly variable. Offsets between sea ice estimates deduced from the abundance of biomarkers and satellite-based sea ice data are attributed to the different time intervals recorded within the sediments and the instrumental records from the study area, which experienced rapid environmental changes during the past 100 years. To distinguish areas characterized by permanently ice-free conditions, seasonal sea ice cover and extended sea ice cover, we apply the concept of the $PIP_{25}$ index from the Arctic Ocean on our data and introduce the term $PIPSO_{25}$ as a potential sea ice proxy. While the trends in $PIPSO_{25}$ are generally consistent with satellite sea ice data and winter sea ice concentrations estimated by diatom transfer functions, more studies on the environmental significance of $IPSO_{25}$ as a Southern Ocean sea ice proxy are needed before this biomarker can be applied for semi-quantitative sea ice reconstructions.

Key Words: biomarker, IPSO$_{25}$, sea ice, Bransfield Strait, satellite observation
**1 Introduction**
In the last century, the Western Antarctic Peninsula (WAP) has undergone a rapid warming of the atmosphere of
$3.7 \pm 1°$ C, which exceeds several times the average global warming (Pachauri et al., 2014; Vaughan et al., 2003).
Simultaneously, a reduction in sea ice coverage (Parkinson and Cavalieri, 2012), a shortening of the sea ice season
(Parkinson, 2002) and a decreasing sea ice extent of ~4-10 % per decade (Liu et al., 2004) are recorded in the
adjacent Bellingshausen Sea. The loss of seasonal sea ice and increased melt water fluxes impact the formation of
deep and intermediate waters, the ocean-atmosphere-exchange of gases and heat, the primary production and
higher trophic levels (Arrigo et al., 1997; Mendes et al., 2013; Morrison et al., 2015; Orsi et al., 2002; Rintoul,
2007). Since the start of satellite-based sea ice observations, however, a slight increase in total Antarctic sea ice
extent has been documented, which contrasts the significant decrease of sea ice in Western Antarctica, especially
around the WAP (Hobbs et al., 2016).
For an improved understanding of the oceanic and atmospheric feedback mechanisms associated with the observed
changes in sea ice coverage, reconstructions of past sea ice conditions in climate sensitive areas such as the WAP
are of increasing importance. A common approach for sea ice reconstructions in the Southern Ocean is based on
the investigation of sea ice associated diatom assemblages preserved in marine sediments (Bárcena et al., 1998;
Gersonde and Zielinski, 2000; Heroy et al., 2008; Leventer, 1998; Minzoni et al., 2015). By means of transfer
functions, this approach can provide quantitative estimates of a paleo sea ice coverage (Crosta et al., 1998; Esper
and Gersonde, 2014a). The application of diatoms for paleoenvironmental studies, however, can be limited by the
selective dissolution of biogenic opal frustules (Burckle and Cooke, 1983; Esper and Gersonde, 2014b) in the
photic zone (Ragueneau et al., 2000) and in surface sediments (Leventer, 1998). As an alternative or additional
approach to diatom studies, Massé et al. (2011) proposed the use of a specific biomarker lipid – a diunsaturated
highly branched isoprenoid alkene (HBI C$_{25:2}$, Fig. 1) – for Southern Ocean sea ice reconstructions. The HBI diene
was first described by Nichols et al. (1988) from sea ice diatoms. [13]C isotopic analyses of the HBI diene suggest a
sea ice origin for this molecule (Sinninghe Damsté et al., 2007; Massé et al., 2011) and this is further corroborated
by the identification of the sea ice diatom *Berkeleya adeliensis* as a producer of this HBI diene (Belt et al., 2016).
*Berkeleya adeliensis* is associated with Antarctic landfast ice and the underlying so-called platelet ice (Riaux-
Gobin and Poulin, 2004). In a survey of surface sediments collected from proximal sites around Antarctica, Belt
et al. (2016) note a widespread sedimentary occurrence of the HBI diene and – by analogy with the Arctic HBI
monoene termed $IP_{25}$ (Belt et al., 2007) – proposed the term $IPSO_{25}$ (Ice Proxy for the Southern Ocean with 25
carbon atoms) as a new name for this biomarker.
In previous studies, an HBI triene (HBI $C_{25:3}$; Fig. 1) found in polar and sub-polar phytoplankton samples (Massé
et al., 2011) has been considered alongside $IPSO_{25}$ and the ratio of $IPSO_{25}$ to this HBI triene hence has been
interpreted as a measure for the relative contribution of organic matter derived from sea ice algae versus open
water phytoplankton (Massé et al., 2011; Collins et al., 2013; Etourneau et al., 2013; Barbara et al., 2013, 2016).
Collins et al. (2013) further suggested that the HBI triene might reflect phytoplankton productivity in marginal ice
zones (MIZ) and, based on the observation of elevated HBI triene concentrations in East Antarctic MIZ surface
waters, this has been strengthened by Smik et al. (2016a). Known source organisms of HBI trienes (Fig. 1 shows
molecular structures of both the E- and Z-isomer) are, for example, *Rhizosolenia* and *Pleurosigma* diatom species
(Belt et al., 2000, 2017). In the subpolar North Atlantic, the HBI Z-triene has been used to further modify the so-
called $PIP_{25}$ index (Smik et al., 2016b) -  an approach for semi-quantitative sea ice estimates. Initially, $PIP_{25}$ was
based on the employment of phytoplankton-derived sterols, such as brassicasterol (24-methylcholesta-5,22E-dien-
3β-ol) and dinosterol (4α,23,24-trimethyl-5α-cholest-22E-en-3β-ol) (Kanazawa et al., 1971; Volkman, 2003), to
serve as open-water counterparts, while $IP_{25}$ reflects the occurrence of a former sea ice cover (Belt et al., 2007;
Müller et al., 2009, 2011). Consideration of these different types of biomarkers helps to discriminate between ice-
free and permanently ice-covered ocean conditions, both resulting in a lack of $IP_{25}$ and $IPSO_{25}$, respectively (for
further details see Belt, 2018; Belt and Müller, 2013). Uncertainties in the source-specificity of brassicasterol
(Volkman, 1986) and its identification in Arctic sea ice samples, however, require caution when pairing this sterol
with a sea ice biomarker lipid for Arctic sea ice reconstructions (Belt et al., 2013). In this context, we note that
Belt et al. (2018) reported that brassicasterol is not evident in the $IPSO_{25}$ producing sea ice diatom *Berkeleya*
*adeliensis*. While the applicability of HBIs (and sterols) to reconstruct past sea ice conditions has been thoroughly
investigated in the Arctic Ocean (Belt, 2018; Stein et al., 2012; Xiao et al., 2015), only two studies document the
distribution of HBIs in Southern Ocean surface sediments (Belt et al., 2016; Massé et al., 2011). The circum-
Antarctic data set published by Belt et al. (2016), however, does neither report HBI triene nor sterol abundances.
Significantly more studies so far focused on the use of $IPSO_{25}$ and the HBI Z-triene for paleo sea ice
reconstructions and these records are commonly compared to micropaleontologial diatom analyses (e.g., Barbara
et al., 2013; Collins et al., 2013; Denis et al., 2010).
Here, we provide a first overview of the distribution of $IPSO_{25}$, HBI trienes, brassicasterol and dinosterol in surface
sediments from the permanently ice-free ocean in the Drake Passage towards the seasonal sea ice inhabited area
of the Bransfield Strait at the northern WAP. Sea ice estimates based on biomarkers are compared to sea ice
concentrations derived from diatom transfer functions and satellite-derived data on the recent sea ice conditions in
the study area. We further introduce and discuss the so-called $PIPSO_{25}$ index (phytoplankton-$IPSO_{25}$ index),
which, following the $PIP_{25}$ approach in the Arctic Ocean (Müller et al., 2011), may serve as a further indicator of
past Southern Ocean sea ice cover.

## 2 Oceanographic setting

The study area includes the southern Drake Passage and the Bransfield Strait located between the South Shetland Islands and the northern tip of the WAP (Fig. 2a and b). The oceanographic setting in the Drake Passage is dominated by the Antarctic Circumpolar Current (ACC) and several oceanic fronts showing large geostrophic water mass flows and subduction and upwelling of water masses (Orsi et al., 1995). The Antarctic Polar Front (APF) divides relatively warm subantarctic waters from the cold and salty Antarctic waters, while the southern Antarctic Circumpolar Current Front (SACCF) often associates with the maximum sea ice extent (Kim and Orsi, 2014). The current system in the Bransfield Strait is relatively complex and the mixture of water masses is not yet well understood (Moffat and Meredith, 2018; Sangrà et al., 2011). A branch of the ACC enters the Bransfield Strait in the west as the Bransfield Current, carrying transitional waters under the influence of the Bellingshausen Sea (Transitional Bellingshausen Sea Water, TBW). The TBW is characterized by a well-stratified, fresh and warm water mass with summer sea surface temperatures (SST) above $0°$ C. Below the shallow TBW, a narrow tongue of circumpolar deep water (CDW) flows along the slope of the South Shetland Islands (Sangrà et al., 2011). In the eastern part, transitional water from the Weddell Sea (Transitional Weddell Sea Water, TWW) enters the Bransfield Strait through the Antarctic Sound and from the Antarctic Peninsula (AP). This water mass corresponds to the Antarctic Coastal Current (Collares et al., 2018; Thompson et al., 2009). The TWW is significantly colder (summer SST $< 0°$ C) and saltier due to extended sea ice formation in the Weddell Sea Gyre. The two water masses are separated at the sea surface by the Peninsula Front characterized by a TBW anticyclonic eddy system (Sangrà et al., 2011). While the TWW occupies the deep water column of the Bransfield Strait (Sangrà et al., 2011), it joins the surface TBW in the southwestern Bransfield Strait (Collares et al., 2018).

Due to high concentrations of dissolved iron on the shelf (Klunder et al., 2014), the area around the WAP is characterized by a high primary production with high vertical export fluxes during early summer associated with the formation of fast sinking mineral aggregates and fecal pellets (Kim et al., 2004; Wefer et al., 1988). The Peninsula Front divides the Bransfield Strait into two biogeographic regimes of high chlorophyll and diatom abundance in the TBW and low chlorophyll values and a pre-dominance of nanoplankton in the TWW (Gonçalves-Araujo et al., 2015), which is also reflected in the geochemistry of surface sediments (Cárdenas et al., 2018).

## 3    Materials and Methods

### 3.1    Sediment Samples and radiocarbon dating

In total, 26 surface sediment samples obtained by multicorers and box corers during the RV *Polarstern* cruise PS97 (Lamy, 2016) were analyzed (Fig. 2, Table 1). All samples were stored frozen and in glass vials. The composition of the sediments ranges from foraminiferal mud in the Drake Passage to diatomaceous mud with varying amounts of ice rafted debris in the Bransfield Strait (Lamy, 2016).

$^{14}$C radiocarbon dating of two samples from the PS97 cruise and one from the *Polarstern* cruise ANT-VI/2 (Fütterer, 1988) was conducted using the mini carbon dating system (MICADAS) at the Alfred Wegener Institute (AWI) in Bremerhaven, Germany, following the method of Wacker et al. (2010). The $^{14}$C ages were calibrated to calendar years before present (cal BP) using the Calib 7.1 software (Stuiver et al., 2019) with an estimated reservoir age of 1178 years, derived from the six closest reference points listed in the Marine Reservoir Correction Database (www.calib.org).

### 3.2    Organic geochemical analyses

For biomarker analyses, sediments were freeze-dried and homogenized using an agate mortar. After freeze-drying, samples were stored frozen to avoid degradation. The extraction, purification and quantification of HBIs and sterols follow the analytical protocol applied by the international community of researchers performing HBI and sterol-based sea ice reconstructions (Belt et al., 2013, 2014; Stein et al., 2012). Prior to extraction, internal standards 7-hexylnonadecane (7-HND) and 5α-androstan-3β-ol were added to the sediments. For the ultrasonic extraction (15 min), a mixture of $CH_2Cl_2$:MeOH (v/v 2:1; 6 ml) was added to the sediment. After centrifugation (2500 rpm for 1 min), the organic solvent layer was decanted. The ultrasonic extraction step was repeated twice. From the combined total organic extract, apolar hydrocarbons were separated via open column chromatography ($SiO_2$) using hexane (5 ml). Sterols were eluted with ethylacetate:hexane (v/v 20:80; 8 ml). HBIs were analyzed using an Agilent 7890B gas chromatography (30 m DB 1MS column, 0.25 mm diameter, 0.250 μm film thickness, oven temperature 60° C for 3 min, rise to 325° C within 23 min, holding 325° C for 16 min) coupled to an Agilent 5977B mass spectrometer (MSD, 70 eV constant ionization potential, ion source temperature 230° C). Sterols were first silylated (200 μl BSTFA; 60° C; 2 hours; Belt et al., 2013; Brault and Simoneit, 1988; Fahl and Stein, 2012) and then analyzed on the same instrument using a different oven temperature program (60° C for 2 min, rise to 150° C within 6 min, rise to 325° C within 56 min 40 sec). As recommended by Belt (2018), the identification of IPSO$_{25}$ and HBI trienes is based on comparison of their mass spectra with published mass spectra (Belt, 2018; Belt et al., 2000; see supplementary material S1). Regarding the potential sulfurization of IPSO$_{25}$, we examined

the GC-MS chromatogram and mass spectra of each sample for the occurrence of the HBI $C_{25}$ sulfide (Sinninghe Damsté et al., 2007). The $C_{25}$ HBI thiane was absent in all samples. For the quantification, manually integrated peak areas of the molecular ions of the HBIs in relation to the fragment ion m/z 266 of 7-HND were used. Instrumental response factors are determined by means of an external standard sediment from the Lancaster Sound, Canada. The HBI concentrations in this sediment are known and a set of calibration series was applied to determine the different response factors of the HBI molecular ions (m/z 346; m/z 348) and the fragment ion of 7-HND (m/z 266) (supplement S2; Belt, 2018; Fahl and Stein, 2012). The identification of sterols was based on comparison of their retention times and mass spectra with those of reference compounds run on the same instrument. Comparison of peak areas of individual analytes and the internal standard was used for sterol quantification. The error determined by duplicate GC-MS measurements was below 0.7 %. The detection limit for HBIs and sterols was 0.5 ng/g sediment. Absolute concentrations of HBIs and sterols were normalized to total organic carbon contents (for TOC data see Cárdenas et al., 2018).

The herein presented phytoplankton-$IPSO_{25}$ index ($PIPSO_{25}$) is calculated using the same formula as for the $PIP_{25}$ index following Müller et al. (2011):

$$PIPSO_{25} = \frac{IPSO_{25}}{IPSO_{25} + (c \times phytoplankton\ marker)} \tag{1}$$

The balance factor c (c = mean $IPSO_{25}$ / mean phytoplankton biomarker) is applied to account for the high offsets in the magnitude of $IPSO_{25}$ and sterol concentrations (see Belt and Müller, 2013; Müller et al., 2011; Smik et al., 2016b for details and a discussion of the c-factor). Since the concentrations of $IPSO_{25}$ and both HBI trienes are in the same range, the c-factor has been set to 1 (following Smik et al., 2016b). For the calculation of the sterol-based $PIPSO_{25}$ index using brassicasterol and dinosterol the applied c-factor is 0.0048 and 0.0137, respectively.

Stable carbon isotope composition of $IPSO_{25}$, requiring a minimum of 50 ng carbon, was successfully determined on five samples using GC-irm-MS. The ThermoFisher Scientific Trace GC was equipped with a 30 m Restek Rxi-5 ms column (0.25 mm diameter, 0.25 μm film thickness) and coupled to a Finnigan MAT 252 isotope ratio mass spectrometer via a modified GC/C interface. Combustion of compounds was done under continuous flow in ceramic tubes filled with Ni wires at 1000° C under an oxygen trickle flow. The same GC program as for the HBI identification was used. The calibration was done by comparison to a $CO_2$ reference gas. The values of $\delta^{13}C$ are expressed in per mill (‰) against Vienna PeeDee Belemnite (VPDB) and the mean standard deviation was <0.9 ‰. An external standard mixture was measured every six runs, achieving a long-term mean standard deviation of 0.2‰ and an average accuracy of <0.1 ‰. Stable isotopic composition of neither HBI trienes nor sterols could be determined due to coeluting compounds.

## 3.3 Diatoms

Details of the standard technique of diatom sample preparation were developed in the micropaleontological laboratory at the Alfred Wegener Institute (AWI) in Bremerhaven, Germany. The preparation included a treatment of the sediment samples with hydrogen peroxide and concentrated hydrochloric acid to remove organic and calcareous remains. After washing the samples several times with purified water, the water was removed and the diatoms were embedded on permanent mounts for counting (see detailed description by Gersonde and Zielinski, 2000). The respective diatom counting was carried out according to Schrader and Gersonde (1978). On average, 400 to 600 diatom valves were counted in each slide using a Zeiss Axioplan 2 at x1000 magnification. In general preservation state of the diatom assemblages was moderate to good in the Bransfield Strait and decreased towards the Drake Passage where it is moderate to poor.

Diatoms were identified to species or species group level and if possible to forma or variety level. The taxonomy follows primarily Hasle and Syvertsen (1996), Zielinski and Gersonde (1997), and Armand and Zielinski (2001). Following Zielinski and Gersonde (1997) and Zielinski et al. (1998) we combined some taxa to groups: The *Thalassionema nitzschioides* group combines *T. nitzschioides* var. *lanceolata* and *T. nitzschioides* var. *capitulata*, two varieties with gradual transition of features between them and no significantly different ecological response. The species *Fragilariopsis curta* and *Fragilariopsis cylindrus* were combined as *F. curta* group taking into account their similar relationship to sea ice and temperature (Armand et al., 2005; Zielinski and Gersonde, 1997). Furthermore, the *Thalassiosira gracilis* group comprises *T. gracilis* var. *gracilis* and *T. gracilis* var. *expecta* because the characteristic patterns in these varieties are often transitional, which hampers distinct identification. Although the two varieties *Eucampia antarctica* var. *recta* and *E. antarctica* var. *antarctica* display different biogeographical distribution (Fryxell and Prasad, 1990), they were combined to the *E. antarctica* group. This group was not included in the transfer function (TF) as it shows no relationship to either sea ice or temperature variation (Esper and Gersonde, 2014a, b). Besides the *E. antarctica* group, we also discarded diatoms assembled as *Chaetoceros* spp. group from the TF-based re-constructions, following Zielinski et al. (1998) and Esper and Gersonde (2014a). This group combines mainly resting spores of a diatom genus with a ubiquitous distribution pattern that cannot be identified to species level due to the lack of morphological features during light microscopic inspection. Therefore, different ecological demands of individual taxa cannot be distinguished.

For estimating winter sea ice (WSI) concentrations we applied the marine diatom TF MAT-D274/28/4an, comprising 274 reference samples from surface sediments in the western Indian, the Atlantic and the Pacific sectors of the Southern Ocean, with 28 diatom taxa and taxa groups, and an average of 4 analogs (Esper and

Gersonde, 2014a). The WSI estimates refer to September sea-ice concentrations averaged over a time period from 1981 to 2010 at each surface sediment site (National Oceanic and Atmospheric Administration, NOAA; Reynolds et al., 2002, 2007). The reference data set is suitable for our approach as it uses a 1° by 1° grid, representing a higher resolution than previously used and results in a root mean squared error of prediction (RMSEP) of 5.52% (Esper and Gersonde, 2014a). We defined 15% concentration as threshold for maximum sea-ice expansion following the approach of Zwally et al. (2002) for the presence or absence of sea ice, and 40% concentration representing the average sea-ice edge (Gersonde et al., 2005; Gloersen et al., 1993). MAT calculations were carried out with the statistical computing software R (R Core Team, 2012) using the additional packages Vegan (Oksanen et al., 2012) and Analogue (Simpson and Oksanen, 2012). Further enhancement of the sea-ice reconstruction was obtained by consideration of the abundance pattern of the diatom sea-ice indicators allowing for qualitative estimate of sea-ice occurrence, as proposed by Gersonde and Zielinski (Gersonde and Zielinski, 2000).

### 3.4 Sea ice data

The mean monthly satellite sea ice concentration was derived from Nimbus-7 SMMR and DMSP SSM/I-SSMIS passive microwave data and downloaded from the National Snow and Ice Data Center (NSIDC; Cavalieri et al., 1996). The sea ice concentration is expressed to range from 0 to 100 %, with concentrations below 15 % suggesting the minor occurrence of sea ice. Accordingly, the sea ice extent is defined as the ocean area with a sea ice cover of at least 15 %.

An interval from 1980 to 2015 was used to generate an average sea ice distribution for each season; spring (SON), summer (DJF), autumn (MAM) and winter (JJA) (Table 2) and the data is considered to reflect the modern mean state of sea ice coverage around the WAP. The high standard deviation in the seasonal sea ice concentrations (up to 26 % in winter; Table 2) in the vicinity of the WAP is attributed to the distinct intra- and interannual variability in sea ice coverage. In this regard, Kim et al. (2005) already related interannual changes in particle flux to annual changes in sea ice cover in the Bransfield Strait. We here suggest that by considering mean sea ice concentrations determined for an observational period of 35 years, reflects a good estimate of average sea ice conditions and facilitates the comparison with sedimentary archives.

## 4    Results and Discussion

In the following we present and discuss the sedimentary concentrations of IPSO$_{25}$, HBI trienes and phytosterols regarding their spatial distribution patterns in relation to the environmental conditions and oceanographic features in the study area. We especially focus on the applicability of these biomarkers for reconstructing sea ice conditions and integrate information derived from satellite observations and diatom-based sea ice estimations. We further discuss the possible approach of a sea ice index PIPSO$_{25}$ by analogy with the Arctic sea ice index PIP$_{25}$ (Müller et al., 2011).

### 4.1    Biomarker distributions in surface sediments

*Distribution of IPSO$_{25}$*

The sea ice biomarker IPSO$_{25}$ was detected in 14 samples, with concentrations ranging between 0.37 and 17.81 μg g$^{-1}$ TOC (Table 1). The distribution of IPSO$_{25}$ in the study area shows a clear northwest-southeast gradient (Fig. 3a) with concentrations increasing from the continental slope and around the South Shetland Islands towards the continental shelf. Maximum IPSO$_{25}$ concentrations are observed at stations under TWW influence with distinctly cold summer SSTs in the Bransfield Strait. According to Belt et al. (2016), deposition of IPSO$_{25}$ is highest in areas covered by landfast sea ice and platelet ice during early spring and summer. Platelet ice is formed under supercooling ocean conditions in the vicinity of ice-shelves and subsequently may be incorporated into drifting sea ice (Gough et al., 2012; Hoppmann et al., 2015). We note that, for example, core sites PS97/068, PS97/069, PS97/072, and PS97/073 in the central and eastern Bransfield Strait are located too distal to be covered by fast ice and suggest that peak IPSO$_{25}$ concentrations at these sites may refer to the frequent drift and melt of sea ice exported from the Weddell Sea into the Bransfield Strait. The vertical export of biogenic material from sea ice towards the seafloor may be accelerated significantly by the formation of organic-mineral aggregates, fecal pellets or by (cryogenic) gypsum ballasting, which promotes a rapid burial and sedimentation of organic matter in polar settings (De La Rocha and Passow, 2007; Wefer et al., 1988; Wollenburg et al., 2018). A recent study from Schmidt et al. (2018) shows that the occurrence of IPSO$_{25}$ in suspended matter and pelagic grazers (krill) is closely linked to the position of the sea ice edge. Lateral subsurface advection of organic matter (including biomarkers) through the TWW, however, may also contribute to elevated IPSO$_{25}$ concentrations at these sites. IPSO$_{25}$ was not detected in sediments from the permanently ice-free areas in the Drake Passage.

The δ$^{13}$C values of IPSO$_{25}$ are between -10.3 ‰ and -14.7 ‰ which is the commonly observed range for IPSO$_{25}$ in surface sediments, sea ice derived organic matter, and in Antarctic krill stomachs (Belt et al., 2016; Massé et al., 2011; Schmidt et al., 2018). These values contrast the low δ$^{13}$C values of marine phytoplankton lipids in

Antarctic sediments (-38 ‰ to -41 ‰ after Massé et al., 2011) and support the sea ice origin of IPSO$_{25}$ in the study
area.
*Distribution of HBI trienes*
The HBI Z-triene was present in all 26 samples (0.33-26.86 µg g$^{-1}$ TOC) and the HBI E-triene was found in 24
samples (0.15-13.87 µg g$^{-1}$ TOC) (Table 1). Highest concentrations of both HBI trienes are found in the eastern
Drake Passage and along the continental slope, where IPSO$_{25}$ is absent, while their concentrations in the Bransfield
Strait are generally low (Fig. 3b and c) suggesting unfavorable environmental conditions for their producers (e.g.,
cooler SSTs, sea ice cover, grazing pressure) for their source diatoms. Contrary to the finding of elevated HBI Z-
triene concentrations in surface waters along an ice-edge (Smik et al., 2016a) and earlier suggestions that this
biomarker may be used as a proxy for MIZ conditions (Belt et al., 2015; Collins et al., 2013; Schmidt et al., 2018),
we observe highest concentrations of the HBI Z- and E-triene at the permanently ice-free northernmost stations
PS97/083 and PS97/084 in the eastern Drake Passage. These core sites are located in close vicinity to the Polar
Front (Fig. 2) and we assume that the productivity of HBI triene source diatoms may benefit from mixing and
upwelling of warm and cold water masses in this area (Moore and Abbott, 2002). Sediments collected south of the
Polar Front and along the Hero Fracture Zone in the western Drake Passage (Fig. 2) contain moderate and very
low concentrations of HBI trienes, respectively. The Hero Fracture Zone is mainly barren of fine-grained
sediments and dominated by sands (Lamy, 2016), which may point to intensive winnowing by ocean currents
impacting the deposition and burial of organic matter. Moderate concentrations of HBI trienes at the continental
slope along the WAP (PS97/053, PS97/074, PS97/077) and in the Bransfield Strait likely refer to primary
production associated with the retreating sea ice margin during spring and summer. This indicates seasonally ice-
free waters in high production coastal areas influenced by upwelling (Gonçalves-Araujo et al., 2015) and feeding
of the local food web (Schmidt et al., 2018). The similarity in the distribution of the HBI Z- and the E-triene in
our surface sediments – the latter of which so far is not often considered for Southern Ocean paleoenvironmental
studies – supports the assumption of a common diatom source for these HBIs (Belt et al., 2000, 2017).
We consider that degradation of biomarker lipids may affect their distribution within surface sediments. While
laboratory studies on HBIs in solution point to a low reactivity of IPSO$_{25}$ towards auto- and photooxidative
degradation (Rontani et al., 2014, 2011), a more recent investigation into Antarctic surface sediments shows that
IPSO$_{25}$ may potentially be affected by partial autoxidative and bacterial degradation but oxidation products are
found in only minor proportions (Rontani et al., 2019a). Since HBI trienes exhibit a generally higher sensitivity to
degradation than the C$_{25}$ HBI diene  (Rontani et al., 2014, 2019b) - and this is supported by a recent observation
of increasing $IPSO_{25}$/HBI triene ratios with increasing water depths in a polynya system off Eastern Antarctica
(Rontani et al., 2019b) – their lower concentrations in the Bransfield Strait have to be considered with care. Vice
versa, regarding maximum HBI triene concentrations and the absence of $IPSO_{25}$ in Drake Passage sediments, we
conclude that the absence of the latter in these samples can be linked to the lack of sea ice (and not to the
degradation of $IPSO_{25}$ as HBI trienes would have been removed first).
*Distribution of sterols*
Brassicasterol is present in all samples with concentrations ranging from 3.39 to 5017.44 µg $g^{-1}$ TOC, while
dinosterol was detected in 22 samples (0.0002-1983.75 µg $g^{-1}$ TOC). It is noticeable that the concentrations of
sterols exceed the concentrations of $IPSO_{25}$ and HBI trienes by more than two orders of magnitude. We observe
higher concentrations of brassicasterol and dinosterol in the eastern part of the Drake Passage supporting an open
marine source for these sterols. Surprisingly, elevated concentrations of brassicasterol are also found at stations
PS97/048-1 and 049-2 in the Hero Fracture Zone, which may argue against a winnowing signal leading to lower
accumulation of organic matter. We can only speculate if transport and deposition of reworked sediment
containing brassicasterol via iceberg rafting could explain these higher values. In contrast to the observation made
for HBI trienes, high sterol concentrations are found in the eastern and central Bransfield Strait (Fig. 3d and e).
Previously, elevated concentrations of steroidal components including brassicasterol and dinosterol in sediment
cores from the Bransfield Strait have been interpreted to reflect a high productivity and significant inputs from
diatoms and dinoflagellates (Brault and Simoneit, 1988). In a more recent overview, also Cárdenas et al. (2018)
report peak concentrations of pigments, sterols and total organic carbon in the Bransfield Strait, which they relate
to large seasonal phytoplankton blooms and higher accumulation rates. Dinosterol and, in particular, brassicasterol
are known to have different source organisms including diatoms, dinoflagellates, cryptophytes, prymnesiophycean
algae and cyanobacteria (Volkman, 1986) and we assume that this diversity accounts for the higher concentration
of these lipids in Bransfield Strait sediments, while concentrations of HBI trienes, mainly derived from diatoms,
are significantly lower. Regarding the potential input of brassicasterol from cryptophytes (Gladu et al., 1990; Goad
et al., 1983), changes in the dominance of this phytoplankton group over diatoms have been reported for our study
area and have been associated with a shallowing of the mixed layer and lower salinity due to intensified glacial
ice-melting along the WAP (Mendes et al., 2013).
Similar to the observations made for HBIs, selective degradation may also affect the concentration of phytosterols
within surface sediments. With respect to the preservation potential of terrigenous and marine derived sterols,
Rontani et al. (2012) note an only weak effect of biotic and abiotic degradation of brassicasterol in Arctic Ocean
shelf sediments – if this is also true for Southern Ocean shelf areas needs to be determined. In general, further
investigations into degradation processes affecting both HBIs and phytosterols within (the same) sediment samples
would address an important knowledge gap regarding in-situ biochemical modifications of the biomarker signal.
**4.2    Comparison of satellite-derived modern sea ice conditions and biomarker data**
The spring and winter sea ice concentrations are shown in Figure 4a and b. Winter sea ice is estimated to not
extend north of 61° S (Fig. 4b) and varies between 1 % and 50 % in the study area, while sea ice is reduced to less
than 20 % in spring (Fig. 4a, Table 2). Sea ice concentrations of up to 50 % are common in winter between the
South Shetland Islands and north of the Antarctic Sound where the influence of TWW is highest. Permanent sea
ice cover is uncommon in the Bransfield Strait and around the WAP and this area is mainly characterized by a
high sea ice seasonality, drift ice from the Weddell Sea (Collares et al., 2018) and a seasonally fluctuating sea ice
margin.
Comparisons of IPSO$_{25}$ and winter sea ice concentrations derived from satellite data reveal a positive correlation
($r^2 = 0.53$). The strongest relationship is observed in the eastern Bransfield Strait where the influence of TWW is
high. Correlations with spring sea ice ($r^2 = 0.27$) and other seasons are weak. As photosynthesis is not possible
and a release of sea ice diatoms from melting sea ice is highly reduced during the Antarctic winter, the observation
of a stronger correlation between recent winter sea ice concentrations and IPSO$_{25}$ is unexpected. We hence suggest
that this offset may be related to the fact that the sediment samples integrate a longer time interval than is covered
by satellite observations. Radiocarbon dating of selected samples that contain calcareous material reveals an age
of 100 years BP in the vicinity of the South Shetland Islands (station PS97/059-2) and 142 years BP at the Antarctic
Sound (station PS1546-2, Table 3). A significantly older age was determined for a sample of *N. pachyderma* from
station PS97/044-1 (4830 years BP) which likely denotes the winnowing and/or very low sedimentation rates in
the Drake Passage. Bioturbation effects and uncertainties in reservoir ages potentially mask the ages of the near-
coastal samples. Nevertheless, since also other published ages of surface sediments within the Bransfield Strait
(Barbara et al., 2013; Barnard et al., 2014; Etourneau et al., 2013; Heroy et al., 2008) are in the range of 0-270
years, we consider that our surface samples likely reflect the paleoenvironmental conditions that prevailed during
the last two centuries (and not just the last 35 years covered by satellite observations). In the context of the rapid
warming during the last century (Vaughan et al., 2003) and the decrease of sea ice at the WAP (King, 2014; King
and Harangozo, 1998), we suggest that the biomarker data of the surface sediments relate to a spring sea ice cover,
which must have been enhanced compared to the recent (past 35 years) spring sea ice recorded via remote sensing.
Presumably, the average spring sea ice conditions over the past 200 years might have been similar to the modern
(past 35 years) winter conditions, which would explain the stronger correlation between IPSO$_{25}$ and winter sea ice
concentrations. The absence of IPSO$_{25}$ at stations PS97/052 and PS97/053, off the continental slope, is in conflict
with the satellite data depicting an average winter sea ice cover of 23 %. Earlier documentations that the IPSO$_{25}$
producing sea ice diatom *Berkeleya adeliensis* favors land-fast ice communities in East Antarctica and platelet ice
occurring mainly in near-coastal areas (Belt et al., 2016; Riaux-Gobin and Poulin, 2004) could explain this
mismatch between biomarker and satellite data, which further strengthens the hypothesis that the application of
IPSO$_{25}$ seems to be confined to continental shelf or near-coastal and meltwater affected environments (Belt, 2018;
Belt et al., 2016). Alternatively, strong ocean currents (i.e. the ACC) could have impacted the deposition of IPSO$_{25}$
in this region.
Although the distribution pattern of HBI trienes reveals generally higher concentrations in ice-free environments,
we note only very weak negative correlations with satellite sea ice data ($r^2 < 0.1$). This may relate to the strong
spatial variability in HBI triene concentrations within the Drake Passage and the different time periods represented
by the satellite and sediment data. Similar to the HBI trienes, also the sterols do not show any significant
relationship to the satellite sea ice concentrations. High abundances of brassicasterol and dinosterol are observed
in both ice free as well as in seasonally ice-covered regions, which points to a broad environmental adaptation of
the source organisms. We hence consider that other environmental parameters than sea ice (e.g., nutrient
availability, water temperature and/or grazing pressure) exert a major control on the productivity of HBI triene
and sterol producers in the study area.
**4.3    Comparison of biomarker distributions and diatom-based sea ice estimates**
The diatoms preserved in sediments from the study area (Table 4) can be associated with open ocean and sea ice
conditions (Fig. 5a-d). North of the South Shetland Islands, the strong influence of the ACC is reflected in the high
abundance of open ocean diatom species such as *Fragilariopsis kerguelensis* and *Thalassiosira lentiginosa* (Esper
et al., 2010). The two diatom species *Fragilariopsis curta* and *Fragilariopsis cylindrus* – known to not produce
HBIs (Belt et al., 2016; Sinninghe Damsté et al., 2004) - mark the vicinity to sea ice (Buffen et al., 2007; Pike et
al., 2008) and indicate fast and melting ice, a stable sea ice margin and stratification due to melting processes and
the occurrence of seasonal sea ice. These observations are in accordance with previous diatom studies revealing a
dominance of *Fragilariopsis kerguelensis* in the permanently open-ocean zone in the Drake Passage and an
assemblage shift to more cold water adapted and sea ice-associated species in the seasonal sea ice zone of the
Bransfield Strait (Cárdenas et al., 2018).
The high abundance of these sea ice diatoms in our samples is in good agreement with high and moderate IPSO$_{25}$
concentrations in the Bransfield Strait and around the South Shetland Islands, respectively. The only HBI source
diatom identified is the HBI Z-triene producing *Rhizosolenia hebetata* (Belt et al., 2017), which is present in four
samples in relatively small amounts which do not show a relation to the measured HBI Z-triene concentrations
(Table 1 and 4). The source diatom of IPSO$_{25}$ *Berkeleya adeliensis* was not observed (or preserved) in the samples,
and we suggest that additional, hitherto unknown, producers for IPSO$_{25}$ as well as for the HBI trienes may exist.
We applied the transfer function of Esper and Gersonde (2014a) with four analogs (4an, Table 4) to our samples
to estimate winter sea ice concentrations (WSI; Figure 5e). The diatom approach shows a clear trend of high winter
sea ice concentrations in the range of 78-91 % in the Bransfield Strait and low sea ice concentrations (between 6-
39 %) north of the continental slope. The fact that diatom data propose sea ice in the Drake Passage may result
from the high ages of surface sediments but also from drift, resuspension and sedimentation of diatom remains.
Because of the absence of IPSO$_{25}$ in the Drake Passage the correlation of its concentrations with WSI is only weak
($r^2 = 0.29$).
**4.4    Testing a semi-quantitative sea ice approach for the Southern Ocean: PIPSO$_{25}$**
Following the PIP$_{25}$-approach applied in the Arctic Ocean (Müller et al., 2011; Belt and Müller, 2013; Xiao et al.,
2015), we used IPSO$_{25}$, HBI triene and sterol data to calculate the PIPSO$_{25}$ index. The main concept of combining
the sea ice proxy with an indicator of an ice-free ocean environment (i.e. a phytoplankton biomarker; Müller et al.,
2011), aims at a more detailed assessment of the sea ice conditions. By reducing the light penetration through the
ice, a thick and perennial sea ice cover limits the productivity of bottom sea ice algae (Hancke et al., 2018), which
results in the absence of both sea ice and pelagic phytoplankton biomarker lipids in the underlying sediments. Vice
versa, sediments from permanently ice-free ocean areas only lack the sea ice biomarker but contain variable
concentrations of phytoplankton biomarkers (Müller et al., 2011). The co-occurrence of both biomarkers in a
sediment sample suggests seasonal sea ice coverage promoting algal production indicative of sea ice as well as
open ocean environments (Müller et al., 2011). Consideration of a phytoplankton biomarker alongside the sea ice
proxy hence helps to avoid an underestimation of the past sea ice cover deduced from the absence of the sea ice
proxy, which, in fact, may also be due to a permanent sea ice cover (Belt, 2018, 2019; Belt and Müller, 2013).
Depending on the biomarker reflecting pelagic (open ocean) conditions, we here define P$_Z$IPSO$_{25}$ (using the HBI
Z-triene), P$_E$IPSO$_{25}$ (using the HBI E-triene), P$_B$IPSO$_{25}$ (using brassicasterol), and P$_D$IPSO$_{25}$ (using dinosterol).
The PIPSO$_{25}$ values are 0 in the Drake Passage and increase to intermediate values at the South Shetland Islands
and the continental slope and reach highest values in the Bransfield Strait (Fig. 6a-d). Minimum PIPSO$_{25}$ values

are supposed to refer to a predominantly ice-free oceanic environment in the Drake Passage, while moderate PIPSO$_{25}$ values mark the transition towards a marginal sea ice coverage at the continental slope and around the South Shetland Islands. Elevated PIPSO$_{25}$ values in samples from the northeastern Bransfield Strait suggest an increased sea ice cover (probably sustained through the drift of sea ice originating in the Weddell Sea). This pattern reflects the oceanographic conditions of a permanently ice-free ocean north of the South Shetland Islands and a seasonal sea ice zone at the WAP influenced by the Weddell Sea as described by Cárdenas et al. (2018). Both HBI triene-based PIPSO$_{25}$ indices show constantly high values at the coast of the WAP of >0.7 (P$_Z$IPSO$_{25}$) and >0.8 (P$_E$IPSO$_{25}$), respectively, and in the southern Bransfield Strait paralleling the southwest-northeast oriented Peninsula Front described by Sangrà et al. (2011). This front is reported to act as a barrier for phytoplankton communities (Gonçalves-Araujo et al., 2015) and is associated with the encounter between TWW carrying Weddell Sea sea ice through the Antarctic Sound and the TBW. The high PIPSO$_{25}$ values suggesting an extended sea ice cover west of the Peninsula Front (station PS97/054 and PS97/056) result from minimum concentrations of pelagic biomarkers and moderate concentrations of IPSO$_{25}$. PIPSO$_{25}$ values based on the HBI E-triene are about 0.2 higher compared to P$_Z$IPSO$_{25}$, due to the generally lower concentrations of the HBI E-triene (Table 1).

The sterol-based PIPSO$_{25}$ values display a generally similar pattern as P$_Z$IPSO$_{25}$ and P$_E$IPSO$_{25}$, respectively, and we note a high comparability between the P$_E$IPSO25 and P$_B$IPSO$_{25}$ values ($r^2 = 0.73$). Some differences, however, are observed in the southwestern part of the Bransfield Strait (station PS97/056) where P$_B$IPSO$_{25}$ indicates a lower sea ice cover and in the central Bransfield Strait (stations PS97/068 and PS97/069) where P$_B$IPSO$_{25}$ and P$_D$IPSO$_{25}$ point to only MIZ conditions. Regarding the modern sea ice conditions, the HBI triene-based PIPSO$_{25}$ indices hence seem to reflect the oceanographic conditions within the Bransfield Strait more satisfactorily. It has to be noted that the brassicasterol- or dinosterol-based PIPSO$_{25}$ index links environmental information derived from biomarker lipids belonging to different compound classes (i.e. HBIs and sterols), which have fundamentally different chemical properties. This requires special attention as, for example, selective degradation of one of the compounds may affect the sedimentary concentration of the respective lipids (Rontani et al., 2018). Previous studies linking HBI and sterol-based sea ice reconstructions with satellite-derived or, with respect to downcore paleo studies, paleoclimatic data, however, demonstrate that the climatic/environmental conditions controlling the production of HBIs and sterols seem to exceed the influence of a potential preferential degradation of these biomarkers within the sediments (e.g., Berben et al., 2014; Cabedo-Sanz et al., 2013; Müller et al., 2009, 2012; Müller and Stein, 2014; Stein et al., 2017; Xiao et al., 2015). A comparison of PIP$_{25}$ records determined using brassicasterol and the HBI Z-triene for three sediment cores from the Arctic realm covering the past up to 14.000 years BP (Belt et al., 2015) reveals very similar trends for both versions of the PIP$_{25}$ index in each core, which

may point to, at least, a similar degree of degradation of HBI trienes and sterols through time. More such studies are needed to evaluate the preservation potential of HBIs and sterols in Southern Ocean sediments, especially for down core paleo studies.

Since brassicasterol and dinosterol are highly abundant in both seasonally ice-covered Bransfield Strait sediments as well as in permanently ice-free Drake Passage sediments, their use as an indicator of fully open-marine conditions in the study area is questionable. Elevated concentrations of both sterols in the Bransfield Strait could either point to an additional input of these lipids from melting sea ice (Belt et al., 2013) or a better adaptation of some of their source organisms to cooler and/or ice-affected ocean environments. Production and accumulation of these lipids in (late) summer (i.e. after the sea ice season) has to be considered as well. This observation highlights the need for a better understanding of the source organisms and the mechanisms involved in the synthesis of these sterols. Similarly, more research is needed on the production of $IPSO_{25}$ in Southern Ocean sea ice environments. The source diatom *Berkeleya adeliensis* seems to be restricted to a very unique ice environment. Previous studies documenting the lack of $IPSO_{25}$ in distal though winter sea ice covered areas (e.g., Belt et al., 2016) emphasize this limitation and it has been suggested that $IPSO_{25}$ may be more indicative of the type of sea ice rather than sea ice extent (Belt, 2019), which needs to be considered when targeting at more quantitative sea ice reconstructions using this biomarker.

*Comparison of $PIPSO_{25}$ with satellite sea ice data and diatom sea ice estimations*

The contour lines in Figure 6a-d show the observed extent of 15 %, 30 %, 40 % and 50 % winter sea ice compared to the $PIPSO_{25}$ values. In the northeastern part of the study area, the HBI triene based $PIPSO_{25}$ indices align well with the contour lines of winter sea ice concentrations and depict the gradient from the marginally ice-covered southern Drake Passage towards the intensively ice-covered Weddell Sea. In the southwestern part of the Bransfield Strait, all $PIPSO_{25}$ indices suggest a higher sea ice cover than it is reflected in the satellite data. This may be explained by the transport (and melt) of drift ice through the TWW, joining the TBW at the southwestern Peninsula Front and/or a higher sea ice cover in this area prior to the remote sensing observational period (and prior to the recent WAP warming).

Correlations of $PIPSO_{25}$ values with satellite-derived sea ice concentrations (for spring, summer, autumn and winter) contrast earlier observations made for the $PIP_{25}$ index in the Arctic Ocean, where the closest linear relationship is found mainly with the spring sea ice coverage (i.e. the blooming season of sea ice algae; Müller et al., 2011; Xiao et al., 2015). We observe a remarkably low correlation between $PIPSO_{25}$ values and spring sea ice concentrations of less than 20 % with a coefficient of determination $r^2 = 0.37$ for $P_ZIPSO_{25}$, $r^2 = 0.50$ for $P_EIPSO_{25}$

(Fig. 7a), $r^2 = 0.31$ for $P_BIPSO_{25}$, and $r^2 = 0.34$ for $P_DIPSO_{25}$ (Fig. 7b). The highest correlation is observed between
winter sea ice concentrations and $P_EIPSO_{25}$ ($r^2 = 0.72$), and $P_ZIPSO_{25}$ ($r^2 = 0.65$, Fig. 7c) with a weaker correlation
for the sterol-based $PIPSO_{25}$ values ($P_BIPSO_{25}$: $r^2 = 0.52$; $P_DIPSO_{25}$: $r^2 = 0.44$, Fig. 7d). As discussed above, we
attribute this seemingly conflicting result of a better agreement between biomarker data and winter (instead of
spring) sea ice conditions to the offset in the time intervals reflected in satellite and sediment data. For the
application of the $PIPSO_{25}$ approach, more aspects concerning the physical environmental conditions controlling
the formation of platelet ice, which, at least at this state of research, is regarded as a main source of $IPSO_{25}$ (Belt
et al., 2016) need to be considered. The formation and accumulation of platelet ice in supercooled waters below
landfast sea ice or underneath an ice-shelf (e.g., Gough et al., 2012; Hoppmann et al., 2015) seem to limit the
spatial occurrence of $IPSO_{25}$ and hence the applicability of $PIPSO_{25}$ to coastal environments. However, transport
of supercooled waters away from the coast may lead to platelet ice formation (and colonization of *Berkeleya*
*adeliensis*) in more distal areas (Hoppmann et al., 2015) and also the drift of sea ice (including the underlying
platelet ice) may impact the distribution of $IPSO_{25}$ in Southern Ocean sediments and these processes require further
investigations. Even though $PIPSO_{25}$ values show a stronger relationship to satellite sea ice concentrations than
$IPSO_{25}$ concentrations more insight into the production and sedimentation of the involved biomarker lipids is
needed to develop such a semi-quantitative approach.
With regard to the spatially and temporally variable sea ice extent, Esper and Gersonde (2014a) studied the
response of diatom species to changes in environmental conditions and their response to the non-linear behavior
of sea ice dynamics (Zwally et al., 2002). In contrast to ice free areas or areas of permanent sea ice cover, areas
characterized by the transition from consolidated to unconsolidated sea ice show rapid changes in satellite derived
sea ice concentrations (ranging from 90 % to 15 %) and exhibit a large variability in species composition. To
reflect this curve in sea ice we hence chose a cubic polynomial regression (polynomial of third degree) to determine
the relation between $PIPSO_{25}$ values and satellite data depicting sea ice concentrations of more than 20 %. A
slightly sigmoid-shaped regression line of winter sea ice concentrations and $PIPSO_{25}$ values depicts the non-
linearity of sea ice cover in different sea ice regimes.
A positive correlation is found between WSI concentrations derived from diatoms and the $PIPSO_{25}$ indices based
on HBI trienes ($P_ZIPSO_{25}$ with $r^2 = 0.76$; $P_EIPSO_{25}$ with $r^2 = 0.77$, Fig. 8a). The correlations of sterol-based $PIPSO_{25}$
values with WSI are slightly lower but in the same range ($P_BIPSO_{25}$ with $r^2 = 0.74$; $P_DIPSO_{25}$ with $r^2 = 0.69$, Fig.
8b). A slightly weaker correlation is noted for diatom- and satellite-based winter sea ice concentrations ($r^2 = 0.63$;
Fig. 8c). Overall, the diatom approach indicates higher sea ice concentrations than the satellite data with an offset
of up to 65 %. This may be due to different sources of satellite reference data used for the transfer function or also

due to the fact that the sediment samples integrate a longer time period with a higher sea ice cover than the satellite data (see discussion in section 4.2). Regarding future sea ice reconstructions based on IPSO$_{25}$ and other biomarkers, we note that the simultaneous study of diatom assemblages provides valuable information on the sea surface conditions and may help to avoid misleading interpretation of the biomarker data (Belt, 2019). Vice versa, while diatom-based transfer functions mainly refer to winter sea ice concentrations, the IPSO$_{25}$ (and PIPSO$_{25}$) signal holds critical information on coastal spring/summer sea ice conditions, which are often crucial for ice-shelf (melting) processes. Pairing the micropaleontological and the biomarker approach hence provides for a more comprehensive reconstruction of Southern Ocean sea ice conditions.

## 5 Conclusions

The distribution of the sea ice biomarker $IPSO_{25}$, related HBI trienes and phytosterols as well as diatoms in a suite of surface sediments from the southern Drake Passage and the WAP reflects recent sea surface water characteristics reasonably well. While highest HBI triene concentrations are observed in the permanently open ocean zone of the Drake Passage, they are significantly reduced in the seasonally ice-covered Bransfield Strait. This pattern is reversed for the sea ice proxy $IPSO_{25}$ and in accordance with previous surface sediment analyses revealing a preferential occurrence of this biomarker in near-coastal environments. The distribution of phytosterols points to a broader environmental significance of brassicasterol and dinosterol in terms of ocean temperature and sea ice tolerance, and/or nutrient availability. Following the $PIP_{25}$ approach established for Arctic Ocean sea ice reconstructions, the herein proposed sea ice index $PIPSO_{25}$ indicates seasonal sea ice cover along the coast of the WAP and in the Bransfield Strait, whereas mainly ice-free conditions prevail in the Drake Passage. In general, this pattern is consistent with satellite-derived sea ice data and diatom-based sea ice estimates and we note that the $PIPSO_{25}$ index seems a potential approach towards semi-quantitative sea ice reconstructions in the Southern Ocean. The recent rapid warming in the study area, however, affects the comparability of proxy and satellite data. The fact that the surface sediments integrate a significantly longer time interval than the remote sensing data thwarts attempts to calibrate $PIPSO_{25}$ values against observed sea ice concentrations. Additional data from other circum-Antarctic coastal (and distal) environments and investigations into potential calibration methods are needed to further develop this approach. Importantly, more information is needed on the mechanisms of $IPSO_{25}$ and HBI triene synthesis, transport and preservation within sediments. Despite a generally good agreement between $PIPSO_{25}$-, diatom- and satellite-based sea ice distributions, we note that the basically different sea ice patterns and sea ice varieties in the Southern Ocean and accordingly different mechanisms controlling the $IPSO_{25}$ signal need to be considered carefully, when adapting a (not yet fully validated) semi-quantitative approach initially developed for the Arctic Ocean.

**Data Availability**

All data can be found in this paper and will be available at the open access repository www.pangaea.de (https://doi.pangaea.de/10.1594/PANGAEA.897165).

**Author contributions**

The study was conceived by MV and JM. Data collections and experimental investigations were done by MV together with OE (diatoms), GM (radiocarbon dating), CH (satellite data), and ES (isotope data). MV wrote the manuscript and did the visualizations. KF provided technical support. JM supervised the study. All authors contributed to the interpretation and discussion of the results and the conclusion of this study.

**Competing interests**

None of the authors has a conflict of interest.

**Acknowledgement**

We thank the captain, crew and chief scientist Frank Lamy of RV Polarstern cruise PS97, and the following supporters: Mandy Kiel and Denise Diekstall (technicians), Lester Lembke-Jene (biology, dating), Liz Bonk and Hendrik Grotheer (from MICADAS), Max Mues (sample preparation), Nicoletta Ruggieri (lab support), Walter Luttmer (lab support). Simon Belt is acknowledged for providing the 7-HND internal standard for HBI quantification. We also acknowledge the two anonymous reviewers and the editor for their constructive and detailed comments. Financial support was provided through the Helmholtz Research grant VH-NG-1101.

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

1    **Figures**

a

b

c

3    **Figure 1: The molecular structures of a) IPSO$_{25}$, b) the HBI Z-triene, and c) the HBI E-triene.**

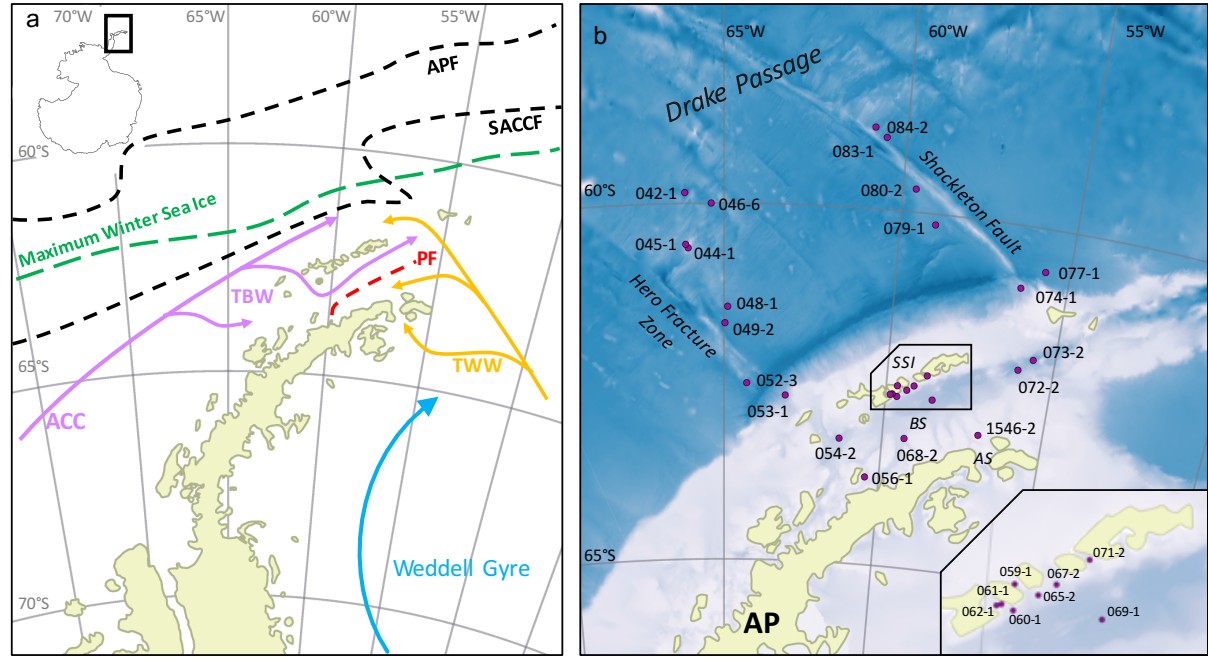

**Figure 2: a) Oceanographic setting of the study area (modified after Hofmann et al., 1996; Sangrà et al., 2011) with**
**ACC = Antarctic Circumpolar Current, TBW = Transitional Bellingshausen Water, TWW = Transitional Weddell**
**Water, APF = Antarctic Polar Front, SACCF = Southern Antarctic Circumpolar Current Front, and PF = Peninsula**
**Front, and the maximum winter sea ice extent (after Cárdenas et al., 2018). b) The bathymetric map of the study area**
**with locations of all stations; AP = Antarctic Peninsula, AS = Antarctic Sound, BS = Bransfield Strait, and SSI = South**
**Shetland Islands. A detailed station map at the South Shetland Islands is integrated.**
**The overview maps were done with QGIS 3.0 from 2018 and the bathymetry was taken from GEBCO_14 from 2015.**

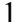

**Figure 3: Distribution of a) IPSO$_{25}$, b) HBI Z-triene, c) HBI E-triene, d) brassicasterol, and e) dinosterol concentrations normalized to TOC. All distribution plots were made with Ocean Data View 4.7.10 (2017).**

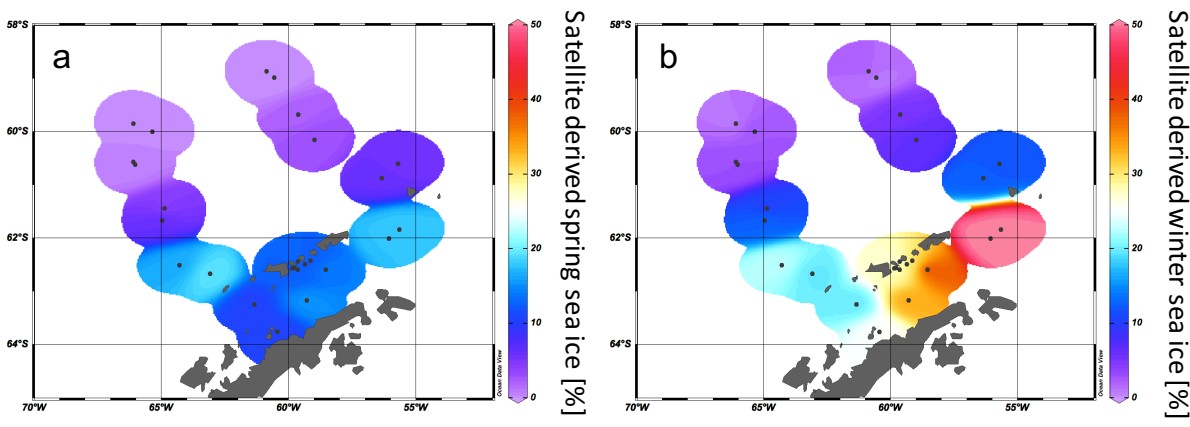

3 **Figure 4: The satellite derived mean sea ice concentrations at each sampling station for a) spring and b) winter.**

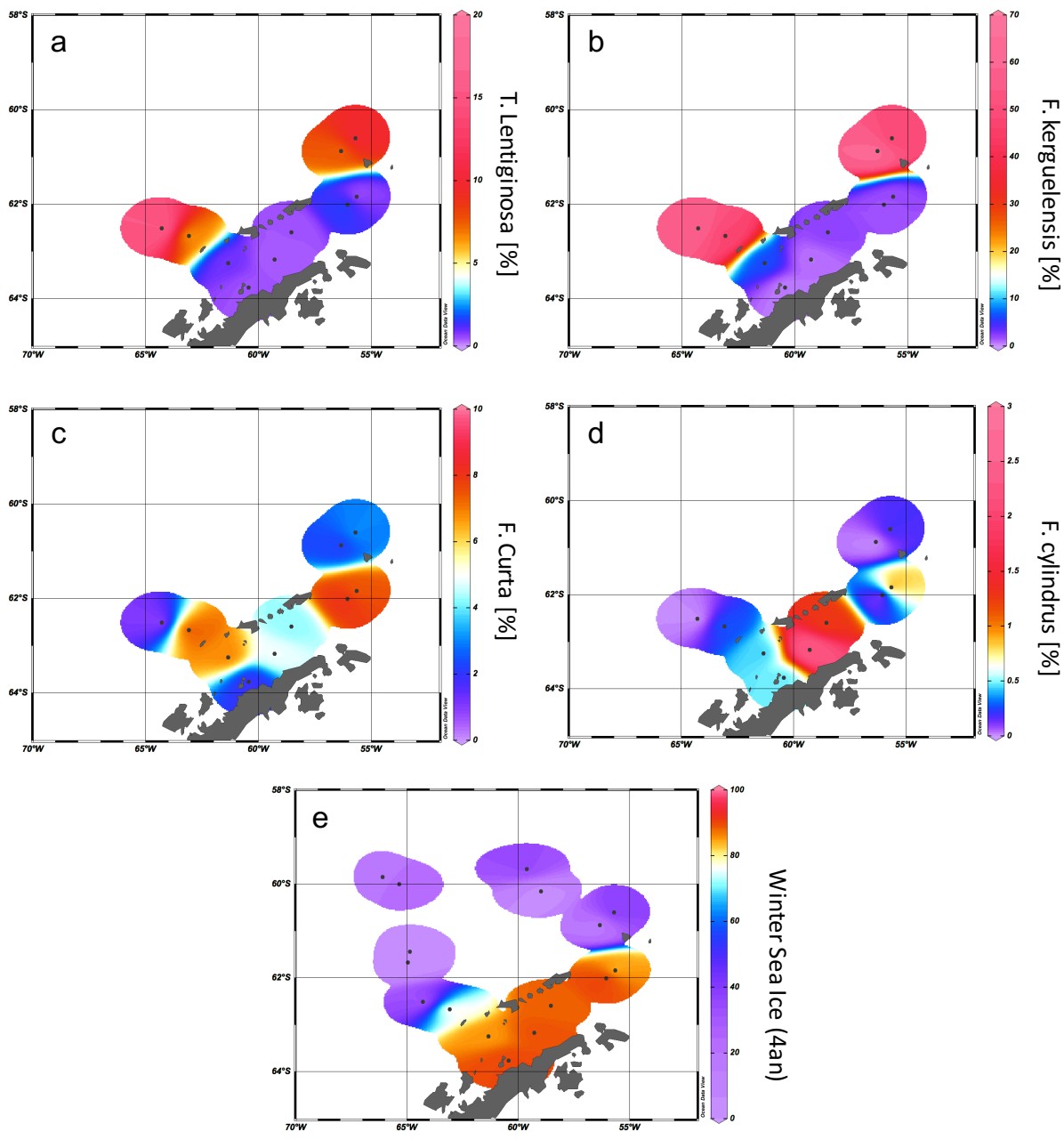

**Figure 5: Distribution of the diatoms a)** *T. lentiginosa***, b)** *F. kerguelensis***, c)** *F. curta***, and d)** *F. cylindrus* **in the study**

**area (percentage per sample). The winter sea ice concentrations from the application of transfer function of Esper and**

**Gersonde** (2014a) **are shown in e).**

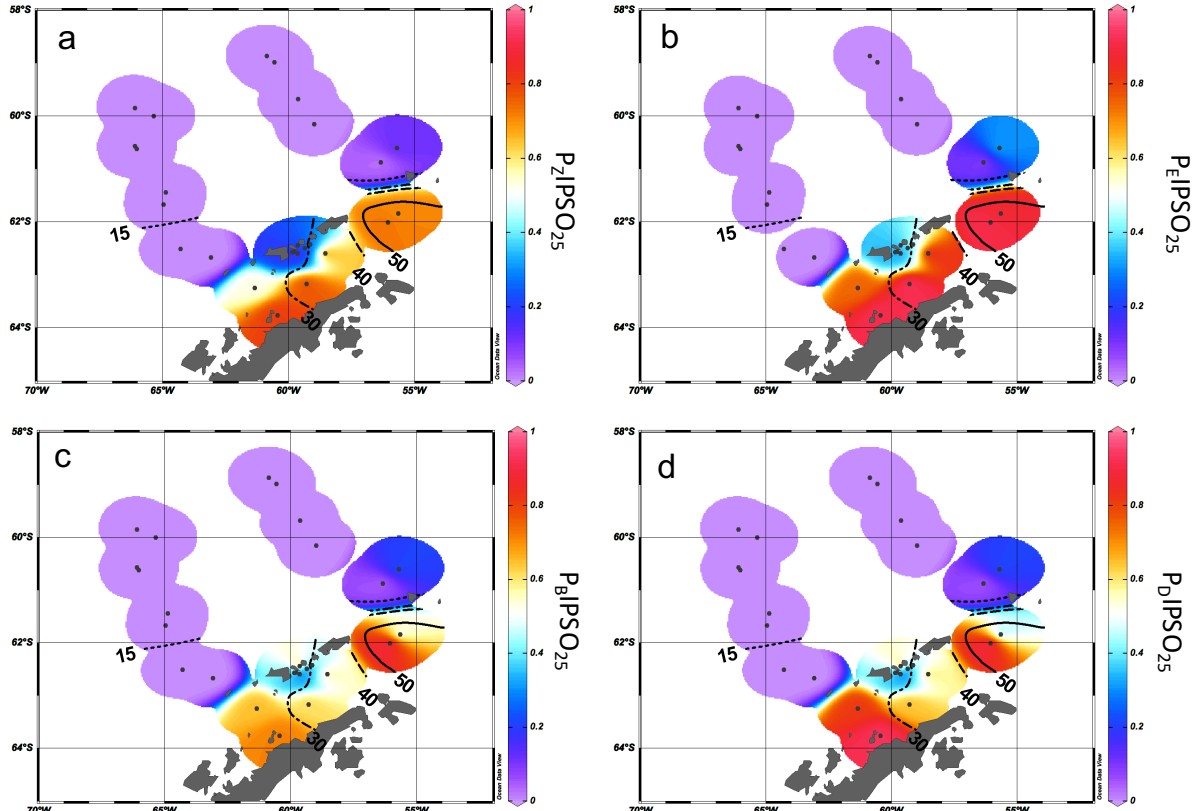

Figure 6: Distribution of a) $P_Z IPSO_{25}$, b) $P_E IPSO_{25}$, c) $P_B IPSO_{25}$, and d) $P_D IPSO_{25}$ values in the study area. The extent of 15 %, 30 %, 40 % and 50 % satellite sea ice concentrations during winter is added as contour lines (see also Figure 4b).

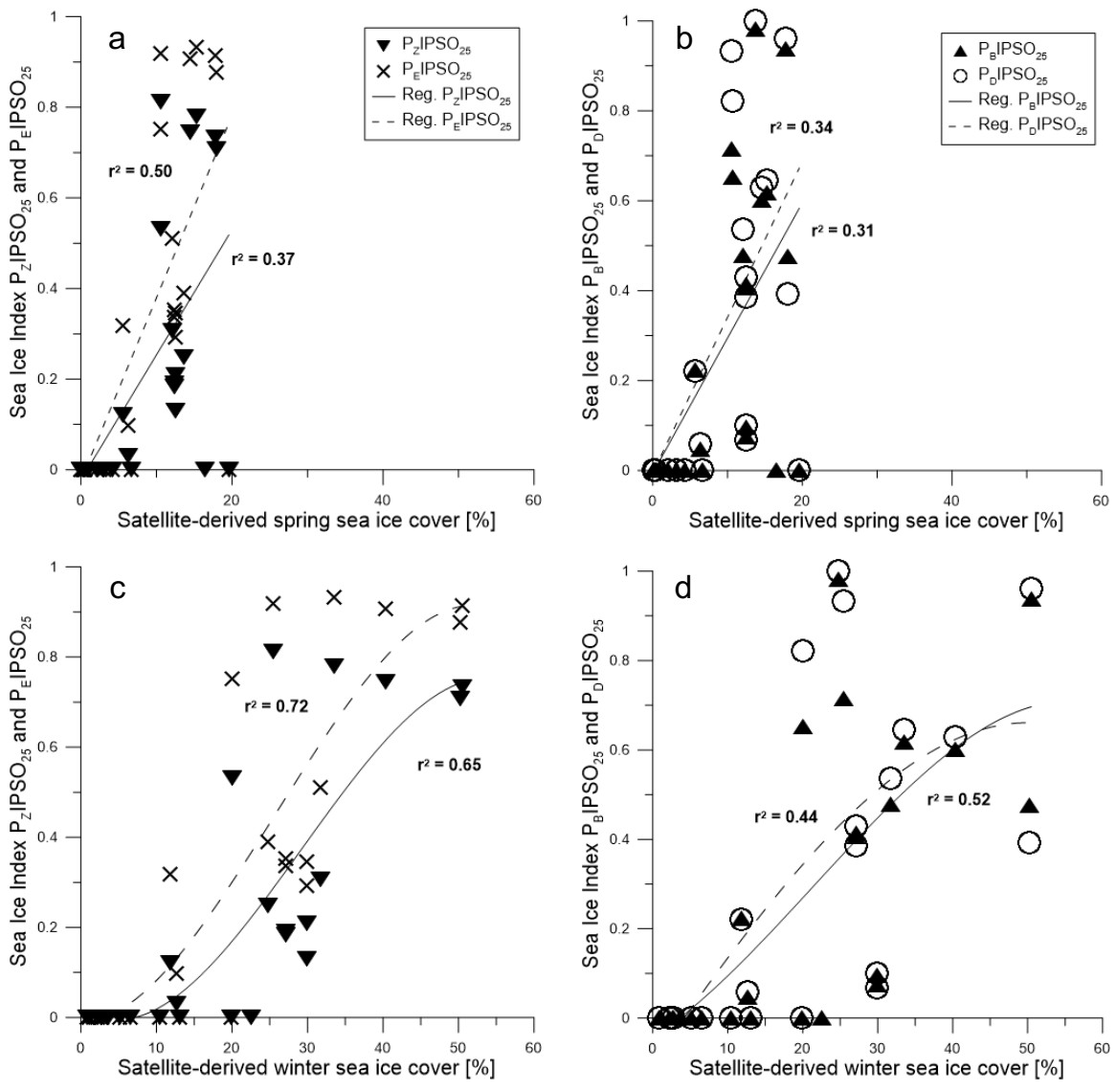

**Figure 7: Scatter plots of satellite spring sea ice concentrations and a) $P_Z IPSO_{25}$ (triangles, solid regression line) and**
**$P_E IPSO_{25}$ (crosses, dashed regression line) and b) $P_B IPSO_{25}$ (triangles, solid regression line) and $P_D IPSO_{25}$ (circles,**
**dashed regression line). Scatter plots of satellite winter sea ice concentrations with c) $P_Z IPSO_{25}$ (triangles, solid**
**regression line) and $P_E IPSO_{25}$ (crosses, dashed regression line) and d) $P_B IPSO_{25}$ (black triangles, solid regression line)**
**and $P_D IPSO_{25}$ (circles, dashed regression line). All scatter plots were done with Grapher™ 13.**

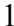

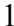

**Figure 8: Scatter plots of a) $P_Z$IPSO$_{25}$ (triangles, solid regression line) and $P_E$IPSO$_{25}$ (crosses, dashed regression line)**
**and b) $P_B$IPSO$_{25}$ (triangles, solid regression line) and $P_D$IPSO$_{25}$ (circles, dashed regression line) against diatom derived**
**winter sea ice concentrations. c) Scatter plot of diatom transfer function winter sea ice concentrations and satellite**
**winter sea ice concentrations.**
**Tables**
**Table 1: Coordinates of sample stations with water depth, concentrations of IPSO$_{25}$, HBI Z- and E-trienes, brassicasterol and dinosterol normalized to TOC, $\delta^{13}$C values for IPSO$_{25}$, and**
**values of sea ice indices PIPSO$_{25}$ based on the HBI Z- and E-trienes, brassicasterol and dinosterol. Concentrations below the detection limit are expressed as 0. The PIPSO$_{25}$ could not be**
**calculated where IPSO$_{25}$ and the phytoplankton marker is absent (blank fields).**

| Station | Lon [dE] | Lat [dN] | Water Depth [m] | IPSO$_{25}$/TOC [µg/g TOC] | HBI Z-Triene /TOC [µg/g TOC] | HBI E-Triene /TOC [µg/g TOC] | Brassicasterol /TOC [µg/g TOC] | Dinosterol /TOC [µg/g TOC] | $\partial^{13}$C of IPSO$_{25}$ [‰] | P$_Z$IPSO$_{25}$ | P$_E$IPSO$_{25}$ | P$_B$IPSO$_{25}$ | P$_D$IPSO$_{25}$ |
|---|---|---|---|---|---|---|---|---|---|---|---|---|---|
| PS97/042-1 | -66.10 | -59.85 | 4172 | 0 | 0.333 | 0.152 | 12.997 | 0 | | 0.000 | 0.000 | 0.000 | |
| PS97/044-1 | -66.03 | -60.62 | 1203 | 0 | 1.080 | 0 | 143.688 | 0 | | 0.000 | | 0.000 | |
| PS97/045-1 | -66.10 | -60.57 | 2292 | 0 | 1.531 | 0.386 | 36.902 | 0 | | 0.000 | 0.000 | 0.000 | |
| PS97/046-6 | -65.36 | -60.00 | 2803 | 0 | 1.359 | 0.291 | 214.634 | 101.809 | | 0.000 | 0.000 | 0.000 | 0.000 |
| PS97/048-1 | -64.89 | -61.44 | 3455 | 0 | 2.085 | 0.375 | 1859.609 | 73.532 | | 0.000 | 0.000 | 0.000 | 0.000 |
| PS97/049-2 | -64.97 | -61.67 | 3752 | 0 | 3.924 | 0.851 | 719.155 | 178.446 | | 0.000 | 0.000 | 0.000 | 0.000 |
| PS97/052-3 | -64.30 | -62.51 | 2890 | 0 | 0.679 | 0 | 26.554 | 0 | | 0.000 | | 0.000 | |
| PS97/053-1 | -63.10 | -62.67 | 2021 | 0 | 19.350 | 5.948 | 13.356 | 332.868 | | 0.000 | 0.000 | 0.000 | 0.000 |
| PS97/054-2 | -61.35 | -63.24 | 1283 | 3.033 | 2.675 | 1.000 | 337.686 | 48.579 | -14.741 | 0.531 | 0.752 | 0.652 | 0.820 |
| PS97/056-1 | -60.45 | -63.76 | 633 | 3.232 | 0.752 | 0.290 | 268.190 | 17.158 | -10.3 ± 0.9 | 0.811 | 0.918 | 0.716 | 0.932 |
| PS97/059-1 | -59.66 | -62.44 | 354 | 0.835 | 2.523 | 1.305 | 3.386 | 0.0002 | | 0.249 | 0.390 | 0.981 | 0.999 |
| PS97/060-1 | -59.65 | -62.59 | 462 | 1.934 | 12.937 | 4.693 | 5017.437 | 1983.750 | | 0.130 | 0.292 | 0.074 | 0.066 |
| PS97/061-1 | -59.80 | -62.56 | 467 | 1.018 | 4.341 | 1.870 | 302.356 | 119.512 | | 0.190 | 0.352 | 0.413 | 0.383 |
| PS97/062-1 | -59.86 | -62.57 | 477 | 0.907 | 4.044 | 1.787 | 276.372 | 88.272 | | 0.183 | 0.337 | 0.407 | 0.428 |
| PS97/065-2 | -59.36 | -62.49 | 480 | 2.416 | 9.184 | 4.549 | 4788.292 | 1587.309 | | 0.208 | 0.347 | 0.095 | 0.100 |
| PS97/067-2 | -59.15 | -62.42 | 793 | 1.785 | 4.038 | 1.710 | 406.567 | 113.728 | | 0.307 | 0.511 | 0.478 | 0.533 |
| PS97/068-2 | -59.30 | -63.17 | 794 | 16.206 | 4.558 | 1.152 | 2096.690 | 653.977 | -14.1 ± 0.6 | 0.780 | 0.934 | 0.617 | 0.643 |

| | | | | | | | | | | | | |
|---|---|---|---|---|---|---|---|---|---|---|---|---|
| PS97/069-1 | -58.55 | -62.59 | 1642 | 17.814 | 6.115 | 1.824 | 2472.025 | 774.345 | -12.6 ± 0.4 | 0.744 | 0.907 | 0.601 | 0.626 |
| PS97/072-2 | -56.07 | -62.01 | 1992 | 13.689 | 4.997 | 1.277 | 192.625 | 40.686 | -13.6 ± 0.3 | 0.733 | 0.915 | 0.937 | 0.961 |
| PS97/073-2 | -55.66 | -61.84 | 2624 | 10.369 | 4.283 | 1.451 | 2388.458 | 1180.752 | | 0.708 | 0.877 | 0.476 | 0.390 |
| PS97/074-1 | -56.35 | -60.87 | 1831 | 0.371 | 12.075 | 3.409 | 1539.629 | 438.073 | | 0.030 | 0.098 | 0.048 | 0.058 |
| PS97/077-1 | -55.71 | -60.60 | 3587 | 2.267 | 16.356 | 4.874 | 1647.616 | 589.731 | | 0.122 | 0.317 | 0.223 | 0.219 |
| PS97/079-1 | -59.00 | -60.15 | 3539 | 0 | 1.893 | 0.510 | 479.917 | 154.400 | | 0.000 | 0.000 | 0.000 | 0.000 |
| PS97/080-2 | -59.64 | -59.68 | 3113 | 0 | 12.021 | 2.705 | 4019.003 | 1329.129 | | 0.000 | 0.000 | 0.000 | 0.000 |
| PS97/083-1 | -60.57 | -58.99 | 3756 | 0 | 18.256 | 8.280 | 686.502 | 308.610 | | 0.000 | 0.000 | 0.000 | 0.000 |
| PS97/084-2 | -60.88 | -58.87 | 3617 | 0 | 26.857 | 13.871 | 1245.652 | 648.474 | | 0.000 | 0.000 | 0.000 | 0.000 |

**Table 2: Seasonal sea ice concentrations from satellite observations for spring, summer, autumn and winter with**

 **standard deviations.**

| Station | Sea Ice Spring [%] | Sea Ice Spring StDev [%] | Sea Ice Summer [%] | Sea Ice Summer StDev [%] | Sea Ice Autumn [%] | Sea Ice Autumn StDev [%] | Sea Ice Winter [%] | Sea Ice Winter StDev [%] |
|---|---|---|---|---|---|---|---|---|
| PS97/042-1 | 0.04 | 0.19 | 0.00 | 0.00 | 0.01 | 0.05 | 1.14 | 5.00 |
| PS97/044-1 | 0.92 | 3.25 | 0.02 | 0.23 | 0.00 | 0.00 | 3.67 | 9.38 |
| PS97/045-1 | 0.52 | 2.08 | 0.01 | 0.08 | 0.00 | 0.04 | 2.65 | 7.81 |
| PS97/046-6 | 0.29 | 1.35 | 0.00 | 0.00 | 0.00 | 0.00 | 2.84 | 8.55 |
| PS97/048-1 | 4.22 | 8.52 | 0.00 | 0.00 | 0.00 | 0.00 | 10.36 | 18.17 |
| PS97/049-2 | 6.65 | 11.85 | 0.00 | 0.00 | 0.00 | 0.04 | 13.02 | 19.91 |
| PS97/052-3 | 16.48 | 21.62 | 0.40 | 2.95 | 0.04 | 0.31 | 22.59 | 24.94 |
| PS97/053-1 | 19.59 | 23.59 | 0.29 | 2.45 | 0.04 | 0.35 | 19.86 | 24.13 |
| PS97/054-2 | 10.62 | 15.18 | 0.44 | 0.79 | 0.76 | 2.62 | 20.06 | 20.72 |
| PS97/056-1 | 10.55 | 16.21 | 4.73 | 3.25 | 2.77 | 4.44 | 25.47 | 23.02 |
| PS97/059-1 | 13.67 | 16.13 | 4.23 | 2.25 | 5.03 | 5.48 | 24.77 | 20.33 |
| PS97/060-1 | 12.53 | 16.84 | 1.87 | 2.15 | 5.43 | 9.24 | 29.93 | 22.05 |
| PS97/061-1 | 12.43 | 16.18 | 1.86 | 2.07 | 4.15 | 7.30 | 27.14 | 21.31 |
| PS97/062-1 | 12.43 | 16.18 | 1.86 | 2.07 | 4.15 | 7.30 | 27.14 | 21.31 |
| PS97/065-2 | 12.53 | 16.84 | 1.87 | 2.15 | 5.43 | 9.24 | 29.93 | 22.05 |
| PS97/067-2 | 12.08 | 17.22 | 0.82 | 1.88 | 5.60 | 10.10 | 31.74 | 22.69 |
| PS97/068-2 | 15.30 | 19.35 | 4.89 | 3.40 | 6.44 | 10.45 | 33.49 | 23.13 |
| PS97/069-1 | 14.51 | 19.85 | 0.40 | 2.34 | 7.83 | 13.78 | 40.41 | 24.27 |
| PS97/072-2 | 17.74 | 22.74 | 1.46 | 5.38 | 16.69 | 20.35 | 50.49 | 25.09 |
| PS97/073-2 | 17.99 | 23.28 | 1.81 | 6.14 | 16.43 | 19.85 | 50.29 | 26.01 |
| PS97/074-1 | 6.30 | 13.65 | 0.02 | 0.12 | 0.55 | 2.29 | 12.65 | 19.30 |
| PS97/077-1 | 5.60 | 12.20 | 0.04 | 0.13 | 0.77 | 2.99 | 11.83 | 17.81 |
| PS97/079-1 | 3.10 | 8.91 | 0.03 | 0.27 | 0.01 | 0.12 | 6.50 | 15.49 |
| PS97/080-2 | 2.08 | 7.52 | 0.01 | 0.08 | 0.00 | 0.04 | 5.14 | 14.17 |
| PS97/083-1 | 0.03 | 0.23 | 0.00 | 0.00 | 0.00 | 0.04 | 0.87 | 4.27 |
| PS97/084-2 | 0.40 | 2.21 | 0.00 | 0.00 | 0.00 | 0.04 | 2.23 | 9.59 |

**Table 3: Details of the radiocarbon dates and calibrated ages.**

| Sample Name | AWI-No. | Material | F14C ± error | Conventional 14C age [a] | Calibrated age (cal BP) [a] |
|---|---|---|---|---|---|
| PS97/044-1 | 1657.1.1 | N. pachyderma | 0.5076 | 5447 ± 111 | 4830 |
| PS97/059-2 | 1434.1.1 | calcareous | 0.8507 | 1299 ± 49 | 100 |
| PS1546-2 | 1602.1.1 | Moll.-Echinod | 0.8456 | 1347 ± 64 | 142 |

18 **Table 4: Estimations of winter sea ice (WSI) derived from diatom species and the distribution of main diatom species in each sample.**

| Station | Diatoms WSI (4an) [%] | A.tabularis [%] | E.antarctica [%] | F.vanheurckii [%] | F.kerguelensis [%] | F.obliquecostata [%] | F.sublinearis [%] | F.curta [%] | F.cylindrus [%] | N.directa [%] | O.weißflogii [%] | P.lineola-turgid.-gr. [%] | R.alata [%] | R.hebetata fo. semispina [%] | S.microtrias [%] | T.lentiginosa [%] | T.oliverana [%] | Thalassiosira MT 3 [%] | P.pseudodenticulata [%] | Stephanopyxis sp. [%] |
|---|---|---|---|---|---|---|---|---|---|---|---|---|---|---|---|---|---|---|---|---|
| PS97/042-1 | 19.2 | 0 | 0 | 0 | 0.8 | 0 | 0 | 0 | 0 | 0 | 0 | 0 | 0 | 0 | 0 | 0 | 0 | 0 | 0 | 0 |
| PS97/046-6 | 24.2 | 0 | 0 | 0 | 0.8 | 0 | 0 | 0 | 0 | 0 | 0 | 0 | 0 | 0 | 0 | 0 | 0 | 0 | 0 | 0 |
| PS97/048-1 | 6.4 | 0 | 0 | 0 | 0.8 | 0 | 0 | 0 | 0 | 0 | 0 | 0 | 0 | 0 | 0 | 0 | 0 | 0 | 0 | 0 |
| PS97/049-2 | 7.7 | 0 | 0 | 0 | 0.7 | 0 | 0 | 0 | 0 | 0 | 0 | 0 | 0 | 0 | 0 | 0 | 0 | 0 | 0 | 0 |
| PS97/052-3 | 32.4 | 1.4 | 4.3 | 0 | 60.3 | 0.2 | 0 | 0.9 | 0 | 0 | 0.5 | 0 | 0 | 0 | 0 | 16.2 | 0.5 | 0 | 0 | 0 |
| PS97/053-1 | 78.1 | 0.4 | 1.2 | 0 | 48.5 | 0.1 | 0.1 | 7.4 | 0.3 | 0 | 0.4 | 0 | 0.3 | 0 | 0.1 | 6.5 | 0 | 0 | 0 | 0 |
| PS97/054-2 | 85.2 | 0 | 0.4 | 0 | 6.2 | 0 | 0 | 6.9 | 0.4 | 0.2 | 2.7 | 0.2 | 0.2 | 0.2 | 0.4 | 0.9 | 0 | 0.2 | 0 | 0.4 |
| PS97/056-1 | 89.9 | 0 | 0 | 0 | 0.5 | 0.3 | 0 | 1.9 | 0.5 | 0.3 | 0.5 | 0 | 0 | 0 | 0.5 | 0.5 | 0 | 0.0 | 0.5 | 0.3 |
| PS97/068-2 | 89.2 | 0 | 0 | 0 | 0.7 | 0.3 | 0.6 | 4.9 | 2.4 | 0.1 | 0.5 | 0.1 | 0 | 0 | 0.4 | 0.4 | 0 | 0.4 | 0.2 | 0 |
| PS97/069-1 | 88.2 | 0 | 0.2 | 0.4 | 2.1 | 0.2 | 0.4 | 4.3 | 1.3 | 0 | 0.6 | 0.2 | 0 | 0 | 0.2 | 0.4 | 0 | 0.2 | 0 | 0 |
| PS97/072-2 | 90.9 | 0 | 0.2 | 1.1 | 1.7 | 0.6 | 0.9 | 8.1 | 0 | 0 | 5.7 | 0.2 | 0.2 | 0 | 0 | 1.7 | 0 | 0.9 | 0.9 | 0 |
| PS97/073-2 | 83.7 | 0 | 0.2 | 0.2 | 1.8 | 0 | 0.4 | 7.4 | 1.0 | 0 | 1.0 | 0 | 1.6 | 0.8 | 0 | 0.4 | 0 | 2.1 | 0.6 | 0 |
| PS97/074-1 | 20.1 | 0.4 | 0.6 | 0 | 63.1 | 0 | 0 | 2.3 | 0 | 0 | 0.2 | 0 | 0.0 | 0.2 | 0 | 7.4 | 0.4 | 0 | 0.2 | 0 |
| PS97/077-1 | 39.4 | 0.6 | 3.3 | 0 | 49.1 | 0.8 | 0 | 3.1 | 0.2 | 0 | 0.2 | 0 | 0.0 | 1.3 | 0 | 10.0 | 0.2 | 0 | 0 | 0 |

| PS97/079-1 | 9.1 | 0 | 0 | 0 | 0.7 | 0 | 0 | 0 | 0 | 0 | 0 | 0 | 0 | 0 | 0 | 0 | 0 | 0 | 0 |
| PS97/080-2 | 35.1 | 0 | 0 | 0 | 0.7 | 0 | 0 | 0 | 0 | 0 | 0 | 0 | 0 | 0 | 0 | 0 | 0 | 0 | 0 |

