# Peer review of "reconstructions: a pilot study from the Western Antarctic"

_Biogeosciences, 2018_

## Referee Comment (RC1) · Anonymous Referee #1 · 8 Feb 2019

The last decade has seen the rise of diatom-specific biomarkers, HBIs, to reconstruct past changes in sea ice. Although the HBI proxy has been studied for its production, seasonal and spatial distribution, export, mineralization and preservation in the peri-Arctic Ocean (Belt, 2018 and references cited therein) much less is known about its Antarctic counterpart. There, very few studies intended to validate the HBIs as a sea ice proxy (Massé et al., 2011; Belt et al., 2016; Smik et al., 2016). Nonetheless, HBIs were applied in several cores to reconstruct sea ice conditions off East Antarctica (Denis et al., 2010; Barbara et al., 2013; Campagne et al., 2015) and the Antarctic

[Figure]

Peninsula (Collins et al., 2013; Etourneau et al., 2013; Barbara et al., 2016), generally in complement to diatom records. The present study represents therefore a timely effort to better understand and validate the HBI proxy. Mimicking the approach done in the northern hemisphere, the present study goes a step beyond previous publication by comparing HBIs to other organic compounds and by providing a semi-quantitative calibration of HBIs vs Antarctic sea ice concentration. However, I have few concerns about the region where the study has been conducted and the subsequent calibration robustness, along with minor concerns, detailed below. Main concerns 1. As stated by the authors, the western Antarctic Peninsula has experienced drastic environmental changes over the past decades with a strong warming and a strong reduction of sea ice cover. It is also known to present a very complexe oceanography due to the interaction between a westerly flow entering Bransfield Strait from the South and an easterly flow entering by the North. The Drake Passage, where half of the surface samples were retrieved from, is also swept by the strong ACC that transports settling particles to the East and winnows surface sediments. As an example, the influence of the AAP-sourced iron to Southern Ocean surface waters is evident until 0° (de Jong et al., 2012) and even diatoms are transported away from their production zone (Crosta et al., 1997). One may question the behavior of organic compounds in these settings. I reckon that authors may have a limited access to sediment material, but the WAP and Drake Passage are not the best regions to calibrate such a tool due to fast changing conditions and strong lateral transport. 2. Fast changing conditions are the hypothesis put forward by the authors to explain the unexpected correlation of PIPSO to winter sea ice, while the IP25 (and probably its IPSO counterpart) is known to be produced in spring (Brown et al., 2011). Here lies the main problem to me. Can we state that the PIPSO preserved in surface sediment relate to past spring sea ice cover and that the correlation to modern winter sea ice cover is because of the close resemblance of past spring sea ice cover to modern winter sea ice cover (page 12, lines 10-19)? Another explanation is that the relationships between PIPSO and spring sea ice and PIPSO and winter sea ice are exactly the same, as shown by figure 5, but that the correlation is

lower for spring sea ice due to a much lower gradient (0-20%) compared to winter sea ice (0-60%) and a greater spread. Additionally, it is worth noting that the relationships between PIPSO and spring/winter sea ice concentration (Fig 5C-D) is evaluated with a simple linear regression while the biplots show an inverse Z with low PIPSO values at low WSIC, a vertical increase (at ~30% for WSIC) and high "quenched" PIPSO values at high sea ice concentration. The spread around the regression line appears very large. 3. Methodological information is lacking. How were stored the freeze-dried samples? In a freezer as recommended? What about potential sulfurization of the HBIs observed in some OSuthern Ocean sediments? What is the detection limit of the GCMS for organic compounds? I think that there is not a single paper on HBIs that discuss analytical reproducibility to account for analytical errors during extraction and measurement on the GCMS. The concentration value itself is calculated via the integration of the MS peak at the organic compound retention time, which is performed via a software. However, from my own knowledge, each sample needs to be visually check for a robust integration, i.e. how the baseline is drawn. Different slopes, double peaks, tailing, etc. . . may induce an error and I always wondered how large can be the error if the baseline is moved a bit. As the PIPSO is a ratio of MS-sourced values, there even might be a propagation of the errors. Can they be quantified? I am also surprised that there is no coherent concentrations between studies and laboratories whereby publications co-signed by Belt present values below 1 $\mu$g/g sediment of IPSO (Belt et al., 2016) while publications co-signed by Massé present values between few to tens of $\mu$g/g sediment for the same regions (references in Belt et al., 2016). As it cannot be related to the nature of the sediment itself (same areas), it might be related to laboratory protocols and analytical differences mentioned above. Although this is beyond the scope of the present paper to deal with general questions, I believe that additional information related to the evaluation of the laboratory protocol used and analytical errors is needed. 4. No information is given on the export and preservation (or potential different degradation) of the organic compounds analyzed here nor on the impact of lateral transport on their distribution. As HBIs and others organic compounds

are produced by different organisms and may be exported at different rates, they may suffer from different degradation rates and variable winnowing, thus altering the PIPSO calculation and the calibration. In conclusion, although the present study is extremely important I believe it fails on its main goal and that it is not possible here to provide a semi-quantitative calibration for sea ice.

Minor concerns 1. Page 2 lines 5: I do not see how relevant are the listed references, especially Anderson et al. 2009. For impact of sea ice meltwater on intermediate and bottom water formation I would check Shin et al., 2003, Rintoul 2007; for heat and gas see Morisson et al. 2015, Holland 2014. 2. Page 2 lines 6: Total Antarctic sea ice increase. 3. Page 2 lines 14-16: Diatom and biogenic silica dissolution mainly occur in the photic zone, not in the sediment (Ragueneau et al., 2000). 4. Page 4. Oceanographic settings are a bit weak. I would recommend to better detail the current. For example, the TBW seems shallower than the TWW, Sangra et al., 2011). What about deeper currents (CDW) than can also winnow particles. What about SST and sea ice cover? Maybe few words on productivity? 5. Page 7, line 1. I would not say that Eucampia antarctica is an open ocean diatom, especially when dealing with its variety recta. Previous works showed that it is mainly affiliated to cold waters and to melting ice (Leventer et al., 2002). So which variety is used here? 6. Page 7, lines 5-8. As far as I know, SMMR is available between 1978 and 1987. Since then, sea ice images are based on SSMI, though several sensors were used over the years, or AMSRE since 2002. 7. Page 8, lines 12-14. Can the presence of PIPSO in central and eastern Bransfield Strait be due to lateral transport? 8. Page 8, lines 20-24. What organisms produce the brassicasterols? How B are exported? It is worth noting that winnowing, due to strong ACC in Drake Passage, can strongly bias surface sediment concentration (focusing, winnowing, etc. . .). For these samples, what can be the impact of lateral transport? 9. Page 9, lines 3-6. d13C was not measured on Triene, Brassicasterol and Dinosterols? Not enough carbon or not necessary? I think that B and D d13C would have be interesting to check whether these compounds were synthesized by sea-ice welling or phytoplankton organisms. Especially as B and D have high concentrations in

Bransfield Strait, similar to IPSO, and we do not know whether this is an autochthone production, and during which season, or an allochthone production transported by the westerly currents. 10. Page 10, line 14. I am not sure that sea ice persists until summer in WAP and BS. At least, low winter sea ice concentrations argue against that. Figure 2 also argue against this. As mentioned above this can be solved by improving the oceanographic settings. 11. Page 10, lines 16-19. This sentence is not totally true. High values of PIPSO are observed at $\sim60°$W in the westernmost BS, where there is no winter sea ice. It should be discussed. 12. Page 11, line 5. Please note that sea ice edge is generally at the 15% threshold. Here 5% means no sea ice. 13. Page 11, lines 24-26. A WSIC of 23% is very low, indicating a very marginal zone. So maybe sensible not to find IPSO but high concentrations of HBI trienes (Figure 3). 14. Page 13, lines 5-8. The interpretation of the biplots is to simplistics. The "good" positive correlations are based on linear regressions that link two patches of samples. There is no gradient. We can imagine intermediate samples out of the linear model. Additionally, there are no t-test for the significance of the correlation. 15. Page 17, references. Check spelling of references. The two first ones mention Science (80-. ).

Comments on figures Figure 2. Add sea ice limits on box B? Caption mentions that the mean WSI and SSI extents were calculated between 1980 and 2010, while it is written 1980 to 2015 in the methods section. Figure 3. There are several ways to present HBIS, ng/g of sediment, normalized to total HBIs, normalized to TOC as in the present study. Are results similar is other options are chosen? Figure 4. I wonder whether that wouldn't be more visual to have colored dots for organic compounds superimposed on WSI concentration fields. Or at least, I would recommend not to interrupt the dash lines for sea ice isocontours. Figure 5. Revise caption whereby plots A-B present spring sea ice concentrations while C-D present winter sea ice concentrations.

---

## Author Comment (AC1) · 22 Feb 2019

We like to thank the reviewer for the very helpful comments. We revised our manuscript taking all concerns into account and uploaded the revised version (see supplement PDF).

Main concerns

1. As stated by the authors, the western Antarctic Peninsula has experienced drastic environmental changes over the past decades with a strong warming and a strong reduction of sea ice cover. It is also known to present a very complex oceanography due to the interaction between a westerly flow entering Bransfield Strait from the South and an easterly flow entering by the North. The Drake Passage, where half of the surface samples were retrieved from, is also swept by the strong ACC that transports settling particles to the East and winnows surface sediments. As an example, the influence of the AAP-sourced iron to Southern Ocean surface waters is evident until 0âŮę(de Jong et al.,2012) and even diatoms are transported away from their production zone (Crosta et al., 1997). One may question the behavior of organic compounds in these settings. I reckon that authors may have a limited access to sediment material, but the WAP and Drake Passage are not the best regions to calibrate such a tool due to fast changing conditions and strong lateral transport.

- We are aware of the difficult and complex hydrographic situation in the study area and already mentioned the potential winnowing of fine grained sediment in the Drake Passage (i.e. the Hero Fracture Zone) in the manuscript. However, we'd like to emphasize that we do not claim to provide a calibration of IPSO25 or the herein proposed PIPSO25 index. Here we document the distribution of HBIs and sterols and introduce the PIPSO25 index as an alternative approach to the commonly applied ratio of IPSO25 against the HBI triene. In fact, lateral transport is an important aspect concerning the deposition and accumulation of organic matter and we now comment on this in the manuscript (see comments below).

2. Fast changing conditions are the hypothesis put forward by the authors to explain the unexpected correlation of PIPSO to winter sea ice, while the IP25 (and probably its IPSO counterpart) is known to be produced in spring (Brown et al., 2011). Here lies the main problem to me. Can we state that the PIPSO preserved in surface sediment relate to past spring sea ice cover and that the correlation to modern winter sea ice cover is because of the close resemblance of past spring sea ice cover to modern winter sea ice cover (page 12, lines 10-19)?

- We formulated this hypothesis due to the better resemblance of PIPSO25 values to the winter sea ice distribution and the 14C ages of surface sediments from the study area.

3. Another explanation is that the relationships between PIPSO and spring sea ice and PIPSO and winter sea ice are exactly the same, as shown by figure 5, but that the correlation is spring sea ice due to a much lower gradient (0-20%) compared to winter sea ice (0-60%) and a greater spread. Additionally, it is worth noting that the relationships between PIPSO and spring/winter sea ice concentration (Fig 5C-D) is evaluated with a simple linear regression while the biplots show an inverse Z with low PIPSO values at low WSIC, a vertical increase (atâĹij30% for WSIC) and high "quenched" PIPSO values at high sea ice concentration. The spread around the regression line appears very large.

- We revised the correlation between the data sets and now consider that changes in sea ice cover and diatom species are non-linear phenomena. So we observe a sigmoid-shaped relationship for the winter sea ice and PIPSO25 values, especially because sea ice estimations between 20% and 80% have a much steeper gradient.

4. Methodological information is lacking. How were stored the freeze-dried samples? In a freezer as recommended? What about potential sulfurization of the HBIs observed in some Southern Ocean sediments? What is the detection limit of the GCMS for organic compounds? I think that there is not a single paper on HBIs that discuss analytical reproducibility to account for analytical errors during extraction and measurement on the GCMS. The concentration value itself is calculated via the integration of the MS peak at the organic compound retention time, which is performed via a software. However, from my own knowledge, each sample needs to be visually check for a robust integration, i.e. how the baseline is drawn. Different slopes, double peaks, tailing, etc...may induce an error and I always wondered how large can be the error if the baseline is moved a bit. As the PIPSO is a ratio of MS-sourced values, there even might be a propagation of the errors. Can they be quantified? I am also surprised that there is no coherent concentrations between studies and laboratories whereby publi-
cations co-signed by Belt present values below 1$\mu$g/g sediment of IPSO (Belt et al., 2016) while publications co-signed by Massé present values between few to tens of $\mu$g/g sediment for the same regions (references in Belt et al., 2016). As it cannot be related to the nature of the sediment itself (same areas), it might be related to laboratory protocols and analytical differences mentioned above. Although this is beyond the scope of the present paper to deal with general questions, I believe that additional information related to the evaluation of the laboratory protocol used and analytical errors is needed.

- We revised the methods section and now refer to the analytical protocols of Belt et al. (2014 – an interlaboratory comparison study) and Stein et al. (2012) and added missing information on the reproducibility of our analyses. Integration of GC-MS peaks is done manually for all compounds (not automatically by the software). We also examined the GC-MS chromatograms of the hydrocarbon fractions of all samples for the C25 HBI sulfide, which may be formed through the intramolecular incorporation of sulfur into a C25:2 HBI alkene (see methods chapter). We cannot comment on the analytical protocol of Massé et al. However, IPSO25 concentrations reported in Belt et al. (2016) vary between 1 and >100 ng/g. IPSO25 concentrations determined for our samples are in the same range (0.6 - 182 ng/g).

5. No information is given on the export and preservation (or potential different degradation) of the organic compounds analyzed here nor on the impact of lateral transport on their distribution. As HBIs and others organic compounds are produced by different organisms and may be exported at different rates, they may suffer from different degradation rates and variable winnowing, thus altering the PIPSO calculation and the calibration. In conclusion, although the present study is extremely important I believe it fails on its main goal and that it is not possible here to provide a semi-quantitative calibration for sea ice.

- We added information about the potential degradation, lateral transport/vertical export and preservation of organic compounds in sections 4.1 and 4.2. (+ improving the

oceanographic setting section). We note that winnowing would affect the biomarker content of the respective surface sediments equally (i.e. winnowing does not lead to a selective removal of specific biomarker compounds – the rate of removal would be the same for all compounds). As already stated in section 4.5, our data set is too small for a full calibration of the PIPSO25 index. As emphasized in the introduction, this study is the first one that provides an overview on the distribution of IPSO25, HBI trienes and sterols in Antarctic marine surface sediments. The combined use of IPSO25 and HBI trienes as well as sterols for a semi-quantitative estimate of the sea ice cover seems promising but a proper calibration requires a larger data set and this is strengthened in the text.

Minor concerns

1. Page 2 lines 5: I do not see how relevant are the listed references, especially Anderson et al. 2009. For impact of sea ice meltwater on intermediate and bottom water formation I would check Shin et al., 2003, Rintoul 2007; for heat and gas see Morisson et al. 2015, Holland 2014.

- We revised the text accordingly.

2. Page 2 lines 6: Total Antarctic sea ice increase.

- We changed this.

3. Page 2 lines 14-16: Diatom and biogenic silica dissolution mainly occur in the photic zone, not in the sediment (Ragueneau et al., 2000).

- We revised this.

4. Page 4. Oceanographic settings are a bit weak. I would recommend to better detail the current. For example, the TBW seems shallower than the TWW, Sangra et al., 2011). What about deeper currents (CDW) than can also winnow particles. What about SST and sea ice cover? Maybe few words on productivity?

[Figure]

- We added information.

5. Page 7, line 1. I would not say that Eucampia antarctica is an open ocean diatom, especially when dealing with its variety recta. Previous works showed that it is mainly affiliated to cold waters and to melting ice (Leventer et al., 2002). So which variety is used here?

- We agree and rephrased this sentence. Because, following Esper and Gersonde (2014), this species is not relevant for the transfer function we deleted this information to avoid confusion.

6. Page 7, lines 5-8. As far as I know, SMMR is available between 1978 and 1987. Since then, sea ice images are based on SSMI, though several sensors were used over the years, or AMSRE since2002.

- We corrected that.

7. Page 8, lines 12-14. Can the presence of PIPSO in central and eastern Bransfield Strait be due to lateral transport?

- We now include the aspect of lateral transport of IPSO25 in the text.

8. Page 8, lines 20-24. What organisms produce the brassicasterols? How B are exported? It is worth noting that winnowing, due to strong ACC in Drake Passage, can strongly bias surface sediment concentration (focusing, winnowing, etc...). For these samples, what can be the impact of lateral transport?

- We now comment on the diverse source organisms for both sterols in the text. We already mentioned that winnowing likely affected the core sites along the Hero Fracture Zone, which lead to the removal of fine-grained sediments (incl. organic compounds). We cannot exclude that winnowing also affected other areas in the Drake Passage. The abundance of fine-grained sediments containing relatively high concentrations of HBI trienes and sterols in the Eastern Drake Passage, however, points to less severe winnowing at these sites. Lateral advection (i.e. additional input) of organic matter may contribute to the biomarker inventory. The absence of IPSO25 at sites in the permanently open ocean zone of the Drake Passage, however, strengthens that the deposition of this lipid is mainly controlled by climate/environmental aspects rather than lateral transport by the ACC.

9. Page 9, lines 3-6. d13C was not measured on Triene, Brassicasterol and Dinosterols? Not enough carbon or not necessary? I think that B and D d13C would have be interesting to check whether these compounds were synthesized by sea-ice welling or phytoplankton organisms. Especially as B and D have high concentrations in Bransfield Strait, similar to IPSO, and we do not know whether this is an autochthone production, and during which season, or an allochthone production transported by the westerly currents.

- We tried to determine the d13C signature of HBI trienes in samples with a sufficiently high concentration of the lipids but co-elution of other compounds hampered the proper peak identification. d13C values obtained for the respective retention time interval were in the range of -30 ‰ (suggesting a phytoplankton source) but the uncertainty in the exact peak allocation prevented us from reporting these values. Due to technical constraints we did not determine d13C of brassicasterol and dinosterol. Regarding the high concentration of the sterols in the Bransfield Strait we now provide further information on their source organisms in the text.

10. Page 10, line 14. I am not sure that sea ice persists until summer in WAP and BS. At least, low winter sea ice concentrations argue against that. Figure2 also argue against this. As mentioned above this can be solved by improving the oceanographic settings.

- We agree with the reviewer and re-phrased the text.

11. Page 10, lines 16-19. This sentence is not totally true. High values of PIPSO are observed atâĹij60âŮęW in the westernmost BS, where there is no winter sea ice. It should be discussed.

- We note a 25 % sea ice cover for this site, which explains the (admittedly low) abundance of IPSO25 in this sample. In the text, we now include a comment on the transport of IPSO25 via the TWW.

12. Page 11, line 5. Please note that sea ice edge is generally at the 15% threshold. Here 5% means no sea ice.

- We revised this part.

13. Page 11, lines24-26. A WSIC of 23% is very low, indicating a very marginal zone. So maybe sensible not to find IPSO but high concentrations of HBI trienes (Figure 3).

- We revised this part.

14. Page 13, lines5-8. The interpretation of the biplots is to simplistics. The "good" positive correlations are based on linear regressions that link two patches of samples. There is no gradient. We can imagine intermediate samples out of the linear model. Additionally, there are no t-test for the significance of the correlation.

- We revised our interpretation of the correlation of the data sets (see major concern no. 2) and chose a sigmoid-shaped regression based on a cubic function for sea ice data covering the transition zone between consolidated and unconsolidated sea ice.

- The t-test on satellite data and diatom based sea ice estimates showed that the correlation is highly significant (p<0.01). Since PIPSO25 values do not reflect the percentage of sea ice cover, a t-test with sea ice concentrations was not performed.

15. Page 17, references. Check spelling of references. The two first ones mention Science (80-. ).

- We corrected this.

Comments on figures Figure 2. Add sea ice limits on box B? Caption mentions that the mean WSI and SSI extents were calculated between 1980 and 2010, while it is written1980 to 2015 in the methods section.

- Adding sea ice limits in box B overcrowds the figure. We revised figure 1 and now indicate the maximum sea ice extent to avoid confusion with the data reported in the main manuscript.

Figure 3. There are several ways to present HBIS, ng/g of sediment, normalized to total HBIs, normalized to TOC as in the present study. Are results similar is other options are chosen?

- Normalizing HBI contents to TOC is common to avoid bias by different sedimentation rates. Plotting HBIs as ng/g sediment and normalized to TOC led to very similar distribution patterns. We further note that normalization to TOC does not affect the PIPSO25 values.

Figure 4. I wonder whether that wouldn't be more visual to have colored dots for organic compounds superimposed on WSI concentration fields. Or at least, I would recommend not to interrupt the dash lines for sea ice isocontours.

- We plotted the data as suggested (WSI in the background, PIPSO25 superimposed) but the visual effect was not satisfying. For clarity, we now provide additional maps showing satellite sea ice concentrations of spring and winter (Fig. 4 e and f).

Figure 5. Revise caption whereby plots A-B present spring sea ice concentrations while C-D present winter sea ice concentrations.

- We revised the caption.

Please also note the supplement to this comment:
https://www.biogeosciences-discuss.net/bg-2018-518/bg-2018-518-AC1-supplement.pdf

**Supplement:**

**Highly branched isoprenoids for Southern Ocean semi-quantitative sea ice reconstructions: a pilot study from the Western Antarctic Peninsula**

[revised manuscript text omitted]

**3.3 Diatoms**

Details of the standard technique of diatom sample preparation were developed in the micropaleontological laboratory at the Alfred Wegener Institute (AWI) in Bremerhaven and described by Gersonde and Zielinski (2000). The counting procedure follows Schrader and Gersonde (1978) with a light microscope Zeiss Axioplan 2 at x1000 magnification.

Since *Chaetoceros* resting spores were highly abundant but not significant for diatom-based environmental analyses applied in this study they were not considered for sea ice calculations. To determine the transfer function for the winter sea ice cover after Esper and Gersonde (2014a), most emphasis was given to the abundance and preservation of the sea ice indicative diatom species *Fragilariopsis curta* and *Fragilariopsis cylindrus*. Hereby, the general preservation state of the diatom assemblages was moderate to good in the Bransfield Strait and decreased towards the Drake Passage where it is moderate to poor. Cold water related diatom species known to dwell in the vicinity of sea ice, like *Thalassiosira weißflogii* and *Porosira pseudodenticulata* (Scott and Thomas, 2005), were also abundant (Table 4).

Diatoms species of *Fragilariopsis kerguelensis* but also A*zpeitia tabularis*, and *Thalassiosira lentiginosa* (and traces of *Thalassiosira oliverana*) were considered to reflect ice free habitats.

**3.4    Sea ice data**

The mean monthly satellite sea ice concentration was derived from Nimbus-7 SMMR and DMSP SSM/I-SSMIS passive microwave data and downloaded from the National Snow and Ice Data Center (NSIDC; Cavalieri et al., 1996).  An interval from 1980 to 2015 was used to generate an average sea ice distribution for each season, spring (SON), summer (DJF), autumn (MAM) and winter (JJA).

**4 Results and Discussion**

**4.1 Distribution of IPSO$_{25}$, HBI trienes and sterols**

The sea ice biomarker IPSO$_{25}$ was detected in 14 samples, with concentrations ranging between 0.37 and 17.81 µg g$^{-1}$ TOC (Table 1). The HBI Z-triene was present in all 26 samples (0.33-26.86 µg g$^{-1}$ TOC) and the HBI E-triene was found in 24 samples (0.15-13.87 µg g$^{-1}$ TOC). Brassicasterol was present in all measured samples with concentrations ranging from 3.39 to 5017.44 µg g$^{-1}$ TOC while dinosterol was detected in 22 samples (0.0002-1983.75 µg g$^{-1}$ TOC).

The distribution of IPSO$_{25}$ in the study area shows a clear northwest-southeast gradient (Fig. 3a) with concentrations increasing from the continental slope and around the South Shetland Islands towards the continental shelf. Maximum IPSO$_{25}$ concentrations are observed in the Bransfield Strait. According to Belt et al. (2016), deposition of IPSO$_{25}$ is highest in area covered by landfast sea ice and platelet ice during early spring and summer. We suppose that core sites PS97/068 to PS97/073 in the central and eastern Bransfield Strait are located too distal to be covered by fast ice and suggest that peak IPSO$_{25}$ concentrations at these sites may refer to the frequent drift and melt of sea ice exported from the Weddell Sea into the Bransfield Strait. The vertical export of biogenic material from sea ice towards the seafloor may be accelerated significantly by the formation of organic-mineral aggregates, fecal pellets or by (cryogenic) gypsum ballasting, which promotes a rapid burial and sedimentation of organic matter in polar settings (De La Rocha and Passow, 2007; Wefer et al., 1988; Wollenburg et al., 2018). Lateral subsurface advection of organic matter (incl. biomarkers) through the TWW, however, may also contribute to elevated IPSO$_{25}$ concentrations at these sites.

IPSO$_{25}$ was not detected in sediments from the permanently ice-free areas in the Drake Passage. Highest concentrations of both HBI trienes are found in the eastern Drake Passage and along the continental slope, while their concentrations in the Bransfield Strait are rather low (Fig. 3b and c) suggesting unfavorable environmental conditions (ocean temperature, sea ice cover) for their source diatoms.

We observe higher concentrations of brassicasterol and dinosterol in the eastern part of the Drake Passage and, in contrast to the observation made for HBI trienes, also in the eastern and central Bransfield Strait (Fig. 3d and e). Dinosterol and, in particular, brassicasterol are known to have different source organisms including diatoms, dinoflagellates, prymnesiophycean algae and cyanobacteria (Volkman, 1986) and we assume that this diversity may account for the higher concentration of these lipids in Bransfield Strait sediments, while concentrations of HBI trienes, mainly derived from diatoms, are significantly lower. Sediments collected along the Hero Fracture Zone in the western Drake Passage (Fig. 2) contain only minor amounts of biomarkers except for elevated brassicasterol concentrations observed at stations PS97/048-1 and 049-2 (Fig. 3d). This part of the Drake Passage is mainly barren of fine-grained sediments and dominated by sands (Lamy, 2016), which may point to intensive winnowing by ocean currents impacting the deposition and burial of organic matter. We consider that also degradation of HBIs and sterols may affect their distribution within surface sediments. Rontani et al. (2014a) report a higher sensitivity of tri-unsaturated HBIs to oxidation but also note that oxidation conditions in pelagic environments (i.e. their source organisms' habitat) are not as significant as those within sea ice. A low reactivity towards oxidative degradation processes is observed for $IPSO_{25}$ (Rontani et al., 2014b, 2011), which supports the good preservation of this lipid in marine sediments. While these degradation studies are commonly conducted on laboratory diatom cultures and phytoplankton cell suspensions, investigations into degradation processes affecting both HBIs and sterols within sediments would address an important knowledge gap regarding in-situ biochemical modifications of the biomarker signal.

The $\delta^{13}C$ values of $IPSO_{25}$ are between -10.3‰ and -14.7‰ which is the commonly observed range for $IPSO_{25}$ in surface sediments and sea ice derived organic matter (Massé et al., 2011, Belt et al., 2016), and contrasts the low

$\delta^{13}C$ values of marine phytoplankton lipids in Antarctic sediments (-38‰ to -41‰ after Massé et al., 2011).

Contrary to the finding of elevated Z-triene concentrations in surface waters along an ice-edge (Smik et al., 2016a)

and earlier suggestions that this biomarker may be used as a proxy for MIZ conditions (Belt et al., 2015; Collins et al., 2013), we observe highest concentrations of the Z- and E-triene at the permanently ice-free northernmost stations in the eastern Drake Passage. This is also apparent for brassicasterol and dinosterol supporting an open marine (pelagic) source for these sterols. Moderate concentrations of HBI trienes at the continental slope along the WAP and in the Bransfield Strait likely refer to primary production at the sea ice margin during spring and summer indicating seasonal ice free waters in high production coastal areas influenced by upwelling (Gonçalves-

[revised manuscript text omitted]

Satellite-derived sea ice data were averaged over the time period from 1980 to 2015 for all four seasons (Table 2)

and are considered to reflect the modern mean state of sea ice coverage around the WAP. The sea ice concentration is expressed to range from 0 to 100 % and, although the error can be up to 15 %, concentrations below 15 % still suggest the occurrence of sea ice. These low sea ice concentrations are usually neglected for the determination of the sea ice extent, which is defined as the ocean area with a sea ice cover of at least 15 %. The spring and winter sea ice concentrations are shown in Figure 4 e-f. Winter sea ice is estimated to not extend north of 61° S (Fig. 4 f)

and varies between 1 % and 50 % in the study area, while sea ice is reduced to less than 20 % in spring (Fig. 4e,

Table 2).

Sea ice concentrations of up to 50 % are common in winter between the South Shetland Islands and north of the Antarctic Sound where the influence of TWW is highest. Permanent sea ice cover is uncommon in the Bransfield Strait and around the WAP and this area is mainly characterized by a high sea ice seasonality and drift ice from the Weddell Sea (Collares et al., 2018). Comparisons of individual biomarker concentrations with satellite sea ice data reveal a weak and positive correlation between IPSO$_{25}$ concentrations and winter sea ice concentrations ($r^2 = 0.5$), while no correlation is found between sea ice and pelagic biomarker concentrations ($r^2 < 0.1$ for all relations). Correlations of PIPSO$_{25}$ values with satellite-derived sea ice concentrations (for spring, summer, autumn and winter) contrast earlier observations made for the PIP$_{25}$ index in the Arctic Ocean, where the closest linear relationship is found mainly with the spring sea ice coverage (i.e. the blooming season of sea ice algae; Müller et al., 2011; Xiao et al., 2015).

Esper and Gersonde (2014a) studied the response of diatom species to changes in environmental conditions and their response to the non-linear behaviour of sea ice dynamics (Zwally et al., 2002). In contrast to ice free areas or areas of permanent sea ice cover, areas characterized by the transition from consolidated to unconsolidated sea ice show rapid changes in satellite derived sea ice concentrations (ranging from 90 % to 15 %) and exhibit a large variability in species composition. To reflect this curve in sea ice we hence chose a cubic polynomial regression (polynomial of third degree) to determine the relation between PIPSO$_{25}$ values and satellite data depicting sea ice concentrations of more than 20 %.

We observe a remarkably low correlation between PIPSO$_{25}$ values and spring sea ice concentrations of less than 20 % with a coefficient of determination $r^2 = 0.37$ for P$_Z$IPSO$_{25}$, $r^2 = 0.50$ for P$_E$IPSO$_{25}$ (Fig. 5a), $r^2 = 0.31$ for P$_B$IPSO$_{25}$, and $r^2 = 0.34$ for P$_D$IPSO$_{25}$ (Fig. 5b). The highest correlation is observed between winter sea ice concentrations and P$_E$IPSO$_{25}$ ($r^2 = 0.72$), and P$_Z$IPSO$_{25}$ ($r^2 = 0.65$, Fig. 5c). A weaker correlation is noted for the sterol-based PIPSO$_{25}$ values (P$_B$IPSO$_{25}$: $r^2 = 0.52$;  P$_D$IPSO$_{25}$: $r^2 = 0.44$, Fig. 5d). The slightly sigmoid-shaped regression line of winter sea ice concentrations and PIPSO$_{25}$ values reflects the non-linearity of sea ice cover in different sea ice regimes as mentioned above.

The contour lines in Figure 4 a-d show the observed extent of 15 %, 30 %, 40 % and 50 % winter sea ice compared to the PIPSO$_{25}$ values. In the northeastern part of the study area, the HBI triene based PIPSO$_{25}$ indices align well with the contour lines of winter sea ice concentrations and depict the gradient from the marginally ice-covered southern Drake Passage towards the intensively ice-covered Weddell Sea. In the southwestern part of the Bransfield Strait, all PIPSO$_{25}$ indices suggest a higher sea ice cover than it is reflected in the satellite data. This may be explained by the transport (and melt) of drift ice through the TWW, joining the TBW at the southwestern Peninsula Front. In contrast, also the absence of IPSO$_{25}$ at stations PS97/052 and PS97/053, off the continental slope, is in conflict with the satellite data depicting an average winter sea ice cover of 23 %. Earlier documentations that the IPSO$_{25}$ producing sea ice diatom *Berkeleya adeliensis* favors land-fast ice communities in East Antarctica (Riaux-Gobin and Poulin, 2004) and platelet ice occurring mainly in near coastal areas (Belt et al., 2016) could explain this mismatch between biomarker and satellite data, which further strengthens the hypothesis that the application of IPSO$_{25}$ seems to be confined to continental shelf or near-coastal and meltwater affected environments (Belt et al., 2016). Alternatively, strong ocean currents (i.e. the ACC) could impact the deposition of IPSO$_{25}$ in this region.

As photosynthesis is not possible and a release of sea ice diatoms from melting sea ice is highly reduced during the Antarctic winter, the observation of a stronger correlation between recent winter sea ice concentrations and PIPSO$_{25}$ sea ice estimates is unexpected. We hence suggest that this offset may be related to the fact that the sediment samples integrate a longer time interval than is covered by satellite observations. Radiocarbon dating of selected samples that contain calcareous material reveals an age of 100 years BP in the vicinity of the South Shetland Islands (station PS97/059-2) and 142 years BP at the Antarctic Sound (station PS1546-2, Table 3). A significantly older age was determined for a sample of *N. pachyderma* from station PS97/044-1 (4830 years BP) in the Drake Passage. Bioturbation effects and uncertainties in reservoir ages potentially mask the ages of the near-coastal samples.

Nevertheless, since also other published ages of surface sediments within the Bransfield Strait (Barbara et al., 2013; Barnard et al., 2014; Etourneau et al., 2013; Heroy et al., 2008) are in the range of 0-270 years, we consider that our surface samples likely reflect the paleoenvironmental conditions that prevailed during the last two centuries (and not just the last 35 years covered by satellite observations). In the context of the rapid warming during the last century (Vaughan et al., 2003) and the decrease of sea ice at the WAP (King, 2014; King and Harangozo, 1998), we suggest that the biomarker data of the surface sediments relate to a spring sea ice cover, which must have been enhanced compared to the recent (past 35 years) spring sea ice recorded via remote sensing. Presumably, the average spring sea ice conditions over the past 200 years might have been similar to the modern (past 35 years) winter conditions, which would explain the stronger correlation between PIPSO$_{25}$ sea ice estimates and winter sea ice concentrations.

**4.4 Comparison of diatom-based sea ice estimates and biomarker data**

The diatoms preserved in sediments from the study area (Table 4) can be associated with open ocean and sea ice conditions. North of the South Shetland Islands, the strong influence of the ACC is reflected in the high abundance of open ocean diatom species such as *Fragilariopsis kerguelensis* and *Thalassiosira lentiginosa* (Esper et al.,

2010). The two diatom species *Fragilariopsis curta* and *Fragilariopsis cylindrus* – known to not produce HBI (Belt et al., 2016; Sinninghe Damsté et al., 2004) – are used for the reconstruction of sea ice conditions (Gersonde and Zielinski, 2000; Xiao et al., 2016). They mark the vicinity to sea ice (Buffen et al., 2007; Pike et al., 2008) and indicate fast and melting ice, a stable sea ice margin and stratification due to melting processes and the occurrence of seasonal sea ice. The high abundance of these species in our samples is in good agreement with high and moderate IPSO$_{25}$ concentrations and PIPSO$_{25}$ values in the Bransfield Strait and around the South Shetland Islands, respectively. The only HBI source diatom identified is the HBI Z-triene producing *Rhizosolenia hebetata* (Belt et al., 2017), which is present in four samples in rather small amounts and does not show a relation to the measured Z-triene concentrations (Table 1 and 4). The source diatom of IPSO$_{25}$ *Berkeleya adeliensis* was not observed (or preserved) in the samples, and we assume that other, hitherto unknown, producers may exist.

We applied the transfer function of Esper and Gersonde (2014a) with four analogs (4an, Table 4) to our samples to compare the different estimates of sea ice cover based on biomarkers and diatoms. A positive correlation of the linear relationship is found between winter sea ice (WSI) concentrations derived from diatoms and the PIPSO$_{25}$ indices based on HBI trienes (P$_Z$IPSO$_{25}$ with $r^2 = 0.76$; P$_E$IPSO$_{25}$ with $r^2 = 0.77$, Fig. 6a). The correlations of sterol-based PIPSO$_{25}$ values with WSI are slightly lower but in the same range (P$_B$IPSO$_{25}$ with $r^2 = 0.74$; P$_D$IPSO$_{25}$ with $r^2 = 0.69$, Fig. 6b). A slightly weaker correlation is noted for diatom- and satellite-based winter sea ice concentrations ($r^2 = 0.63$; Fig. 6c). Overall, the diatom approach indicates higher sea ice concentrations than the satellite data with an offset of up to 65 %. This may be due to different sources of satellite reference data used for the transfer function or also due to the fact that the sediment samples integrate a longer time period with a higher sea ice cover than the satellite data (see discussion in section 4.3).

**4.5 Application of PIPSO$_{25}$ as a semi-quantitative sea ice index**

Precise and, in particular, quantitative reconstructions of past sea ice coverage are crucial for a robust assessment of feedback mechanisms in the ice-ocean-atmosphere system. While diatom transfer functions provide a valuable tool, additional information on sea ice conditions in coastal ice-shelf proximal areas, which are often affected by opal dissolution, are essential. The PIPSO$_{25}$ approach seems to be a promising step into this direction, though our data obtained for the WAP are not yet sufficient for a full calibration. PIPSO$_{25}$, diatom and satellite sea ice data, however, reveal positive correlations (Figs. 5 and 6) and depict similar gradients in sea ice cover. The observed offset between satellite data and biomarker- and diatom-based sea ice estimates likely relates to the fact that the instrumental records cover a significantly shorter or more recent time interval than the studied sediments. The recent rapid warming along the WAP (Vaughan et al., 2003) hence complicates attempts to calibrate these proxy data against observational data. The high correlation between diatom-derived winter sea ice concentrations and

PIPSO$_{25}$ values (Fig. 6a and b) may even argue for a calibration of the IPSO$_{25}$ index against diatom data. The robustness and reliability of such an approach, however, has to be proven by means of a larger data set. Regarding the interpretation of PIPSO$_{25}$ in terms of sea ice coverage in the study area, lower PIPSO$_{25}$ values (<0.15 for

P$_Z$IPSO$_{25}$; <0.31 for P$_E$IPSO$_{25}$; <0.22 for P$_B$IPSO$_{25}$ and P$_D$IPSO$_{25}$) roughly seem to reflect unconsolidated, drifting winter sea ice and a nearly ice-free spring season. Higher values (>0.71 for P$_Z$IPSO$_{25}$; >0.9 P$_E$IPSO$_{25}$; >0.6 for

P$_B$IPSO$_{25}$ and P$_D$IPSO$_{25}$) would refer to an extended winter sea ice cover (up to 91 % in some years) with ice floes remaining until summer.

**5 Conclusion**

[revised manuscript text omitted]

Rontani, J.-F., Belt, S. T., Brown, T. A., Vaultier, F. and Mundy, C. J.: Sequential photo- and autoxidation of diatom lipids in Arctic sea ice, Org. Geochem., 77, 59–71, doi:10.1016/j.orggeochem.2014.09.009, 2014b.

Rontani, J. F., Belt, S. T., Vaultier, F. and Brown, T. A.: Visible light induced photo-oxidation of highly branched isoprenoid (HBI) alkenes: Significant dependence on the number and nature of double bonds, Org. Geochem.,

42(7), 812–822, doi:10.1016/j.orggeochem.2011.04.013, 2011.

Sangrà, P., Gordo, C., Hernández-Arencibia, M., Marrero-Díaz, A., Rodríguez-Santana, A., Stegner, A., Martínez-

Marrero, A., Pelegrí, J. L. and Pichon, T.: The Bransfield current system, Deep Sea Res. Part I Oceanogr. Res.

Pap., 58(4), 390–402, doi:10.1016/J.DSR.2011.01.011, 2011.

Schrader, H. and Gersonde, R.: Diatoms and silicoflagellates, in Micropaleontological Methods and Techniques -

An Excercise on an Eight Meter Section of the Lower Pliocene of Capo Rossello, Sicily, Utrecht

Micropaleontological Bulletins, vol. 17, edited by W. J. Zachariasse, W. R. Riedel, A. Sanfilippo, R. R. Schmidt, M.

J. Brolsma, H. J. Schrader, R. Gersonde, M. M. Drooger, and J. A. Broekman, pp. 129–176., 1978.

Scott, F. J. and Thomas, D. P.: Diatoms, in Antarctic Marine Protists, p. 563, Australian Biological Resources

Study, Canberra & Hobart., 2005.

[revised manuscript text omitted]
/080-2 | 35.1 | 0 | 0 | 0 | 0.7 | 0 | 0 | 0 | 0 | 0 | 0 | 0 | 0 | 0 | 0 | 0 | 0 | 0 | 0 | 0 |

---

## Referee Comment (RC2) · Anonymous Referee #2 · 17 Apr 2019

While a lot of effort has gone into evaluation and calibration of the HBI proxies in the Arctic (their source, environmental factors affecting their production, relation of surface sediment HBI concentrations to sea ice concentrations based on satellite data etc.), very little has been done in the more remote Southern Ocean. The study by Vor- rath et al. is hence a valuable effort to better understand HBI sea ice proxies around Antarctica. The study nicely combines sea ice (related) biomarkers, open water phyto- plankton markers and qualitative and quantitative diatom data from surface sediments, and compares these with satellite-derived sea ice concentrations in the study area.

As reviewer 1 already pointed out, there are some issues about the region where the study has been conducted, relating to the time period the surface sediments represent (especially in the Drake Passage due to winnowing). However, as the authors are out to explore rather than to strictly calibrate their sea ice proxies (especially the PIPSO25 index), this is not a crucial problem and the authors are aware of it. This is in general one of the most challenging issues with calibration of surface sediment proxies against instrumental data, as there rarely is a good control on the time period the surface sediments represent. As a comment on another main point reviewer 1 raised, PIPSO25 is produced in the spring shortly before the ice melts, not in the winter, so the authors' hypothesis about the surface sediments representing a time when spring sea ice cover resembled modern winter sea ice cover appears plausible. Otherwise I agree with the thorough reviews & comments by the editor and reviewer 1, and have only a few more comments/corrections to add:

Page 3, line 3: Initially,.. Page 5, 3.1. It would have been useful to date the surface sediment with 210Pb, to constrain the time period the surface sediments represent. 210Pb is more suited for dating surface sediments than 14C. I am not asking the authors to conduct this now, but it is something to consider for future work. Page 6, 3.3.: Insufficient information is given for diatom methodology. In addition to references given, at least a brief summary of the sample preparation should be provided, whereas the transfer function technique used should be presented with much greater detail! Page 7, line 1: The diatom species Fragilariopsis... Page 9, line 1 – Rontani et al. (2019) Autoxidation of the sea ice biomarker proxy IPSO25 in the near-surface oxic layers of Arctic and Antarctic sediments. Page 9, line 13 ...seasonally ice-free + line 17: ...ice-covered.., + line 27: phytoplankton... Page 9, line 29: remove comma after "both" Page 10, line 4: add comma after 2015) Page 12, lines 7-9: The likelihood of winnowing in Drake Passage should be brought up also here in the discussion. Page 13, lines 25-29: Although an attractive idea – and circumventing the time window issue when calibrating against satellite data - calibration of the PIPSO25 index against diatom data faces issues around the reliability and robustness of diatom-based quantitative inferences,
which should certainly be mentioned!

---

## Author Comment (AC2) · 7 May 2019

Dear reviewers, thank you very much for the very helpful comments. We revised our manuscript under consideration of your comments and uploaded the final, revised version of the manuscript (see supplements). (response to the first reviewers comments can be found in AC1).

Page 3, line 3: Initially,.. - We corrected this.

Page 5, 3.1. It would have been useful to date the surface sediment with 210Pb, to constrain the time period the surface sediments represent. 210Pb is more suited for dating surface sediments than 14C. I am not asking the authors to conduct this now, but it is something to consider for future work. - Unfortunately, we did not have the resources for 210 Pb dating but our colleagues from Chile confirmed recent ages for a single sample (PS97/071-2) in the Bransfield Strait (personal communication).

Page 6, 3.3.: Insufficient information is given for diatom methodology. In addition to references given, at least a brief summary of the sample preparation should be provided, whereas the transfer function technique used should be presented with much greater detail! - We revised the diatom methodology part.

Page7, line 1: The diatom species Fragilariopsis... - We corrected this.

Page 9, line 1 – Rontani et al. (2019) Autoxidation of the sea ice biomarker proxy IPSO25 in the near-surface oxic layers of Arctic and Antarctic sediments. - We included this publication.

Page 9, line 13...seasonally ice-free + line 17:... ice-covered.., + line 27: phytoplankton... - We corrected this.

Page 9, line 29: remove comma after "both" - We corrected this.

Page 10, line 4: add comma after 2015) - We corrected this.

Page 12, lines 7-9: The likelihood of winnowing in Drake Passage should be brought up also here in the discussion. - We added this.

Page 13, lines 25-29: Although an attractive idea – and circumventing the time window issue when calibrating against satellite data - calibration of the PIPSO25 index against diatom data faces issues around the reliability and robustness of diatom-based quantitative inferences, which should certainly be mentioned! - We removed this.

Please also note the supplement to this comment:
https://www.biogeosciences-discuss.net/bg-2018-518/bg-2018-518-AC2- supplement.pdf

**Supplement:**

**Highly branched isoprenoids for Southern Ocean semi-quantitative sea ice reconstructions: a pilot study from the Western Antarctic Peninsula**

[revised manuscript text omitted]

5977B mass spectrometer (MSD, 70 eV constant ionization potential, ion source temperature 230° C). Sterols were first silylated (200 μl BSTFA; 60° C; 2hours) and then analyzed on the same instrument using a different oven temperature program (60° C for 2 min, rise to 150° C within 6 min, rise to 325° C within 56 min 40 sec). As recommended by Belt (2018), the identification of $IPSO_{25}$ and HBI trienes is based on comparison of their mass spectra with published mass spectra (Belt et al., 2000). Regarding the potential sulfurization of $IPSO_{25}$, we examined the GC-MS chromatogram and mass spectra of each sample for the occurrence of the HBI $C_{25}$ sulfide (Sinninghe Damsté et al., 2007). The $C_{25}$ HBI thiane was absent in all samples. For the quantification, manually integrated peak areas of the molecular ions of the HBIs in relation to those of 7-HND were used. An external calibration for HBI diene and trienes was applied using a sample with known HBI concentrations from the Lancaster Sound, Canada, to account for 
[revised manuscript text omitted]
 only minor amounts of biomarkers except for elevated brassicasterol concentrations observed at stations PS97/048-1 and 049-2 (Fig. 3d). This part of the Drake Passage is mainly barren of fine-grained sediments and dominated by sands (Lamy, 2016), which may point to intensive winnowing by ocean currents impacting the deposition and burial of organic matter. We consider that also degradation of HBIs and sterols may affect their distribution within surface sediments. Rontani et al. (2014) report a higher sensitivity of tri-unsaturated HBIs to oxidation but also note that oxidation conditions in pelagic environments (i.e. their source organisms' habitat) are not as significant as those within sea ice. A more recent study by Rontani et al. (2019) on surface sediments shows that $IPSO_{25}$ may potentially be affected by autoxidative and bacterial degradation but oxidation products are found in only minor proportions. In general, further investigations into degradation processes affecting both HBIs and sterols within sediments would address an important knowledge gap regarding in-situ biochemical modifications of the biomarker signal.

The $\delta^{13}C$ values of $IPSO_{25}$ are between -10.3‰ and -14.7‰ which is the commonly observed range for $IPSO_{25}$ in surface sediments and sea ice derived organic matter (Massé et al., 2011, Belt et al., 2016), and contrasts the low

$\delta^{13}C$ values of marine phytoplankton lipids in Antarctic sediments (-38‰ to -41‰ after Massé et al., 2011).

Contrary to the finding of elevated Z-triene concentrations in surface waters along an ice-edge (Smik et al., 2016a)

and earlier suggestions that this biomarker may be used as a proxy for MIZ conditions (Belt et al., 2015; Collins et al., 2013), we observe highest concentrations of the Z- and E-triene at the permanently ice-free northernmost stations in the eastern Drake Passage. This is also apparent for brassicasterol and dinosterol supporting an open marine (pelagic) source for these sterols. Moderate concentrations of HBI trienes at the continental slope along the WAP and in the Bransfield Strait likely refer to primary production at the sea ice margin during spring and summer indicating seasonally ice-free waters in high production coastal areas influenced by upwelling (Gonçalves-

Araujo et al., 2015). The similarity in the distribution of the Z- and the E-triene in our surface sediments – the latter of which so far is not often considered for Southern Ocean paleoenvironmental studies – supports the assumption of a common diatom source for these HBIs (Belt et al., 2000, 2017). Since brassicasterol and dinosterol are highly abundant in both seasonally ice-covered Bransfield Strait sediments as well as in permanently ice-free

Drake Passage sediments, their use as an indicator of fully open-marine conditions is questionable. Elevated concentrations of both sterols in the Bransfield Strait could either point to an additional input of these lipids from melting sea ice (Belt et al., 2013) or a better adaptation of some of their source organisms to cooler and/or ice- dominated ocean conditions. Production and accumulation of these lipids in (late) summer (i.e. after the sea ice season) may be considered as well. This observation highlights the need for a better understanding of the source organisms and the mechanisms involved in the synthesis of these sterols.

**4.2    A novel sea ice index for the Southern Ocean: PIPSO$_{25}$**

The main concept of combining the sea ice proxy with an indicator of an ice-free ocean environment (i.e. a phytoplankton biomarker, Müller et al., 2011), aims at a semi-quantitative assessment of the sea ice conditions.

By reducing the light penetration through the ice, a thick and perennial sea ice cover limits the productivity of bottom sea ice algae (Hancke et al., 2018), which results in the absence of both sea ice and pelagic phytoplankton biomarker lipids in the underlying sediments. Vice versa, sediments from permanently ice-free ocean areas only lack the sea ice biomarker but contain variable concentrations of phytoplankton biomarkers (Müller et al., 2011).

The co-occurrence of both biomarkers in a sediment sample suggests seasonal sea ice coverage promoting algal production indicative of sea ice as well as open ocean environments (Müller et al., 2011).

Following the PIP$_{25}$-approach applied in the Arctic Ocean (Müller et al., 2011; Belt and Müller, 2013; Xiao et al.,

2015), we used IPSO$_{25}$, HBI triene and sterol data to calculate the PIPSO$_{25}$ index. Depending on the biomarker reflecting pelagic (open ocean) conditions, we define P$_Z$IPSO$_{25}$ (using the Z-triene), P$_E$IPSO$_{25}$ (using the E-triene),

P$_B$IPSO$_{25}$ (using brassicasterol), and P$_D$IPSO$_{25}$ (using dinosterol). Since the concentrations of IPSO$_{25}$ and both HBI

trienes are in the same range, the application of the c-factor is not needed here. For the calculation of P$_B$IPSO$_{25}$

the c-factor is 0.0048, for P$_D$IPSO$_{25}$ it is 0.0137.

[revised manuscript text omitted]

Satellite-derived sea ice data were averaged over the time period from 1980 to 2015 for all four seasons (Table 2)

and are considered to reflect the modern mean state of sea ice coverage around the WAP. The sea ice concentration is expressed to range from 0 to 100 % and, although the error can be up to 15 %, concentrations below 15 % still suggest the occurrence of sea ice. These low sea ice concentrations are usually neglected for the determination of the sea ice extent, which is defined as the ocean area with a sea ice cover of at least 15 %. The spring and winter sea ice concentrations are shown in Figure 4 e-f. Winter sea ice is estimated to not extend north of 61° S (Fig. 4 f)

and varies between 1 % and 50 % in the study area, while sea ice is reduced to less than 20 % in spring (Fig. 4e,

Table 2).

Sea ice concentrations of up to 50 % are common in winter between the South Shetland Islands and north of the Antarctic Sound where the influence of TWW is highest. Permanent sea ice cover is uncommon in the Bransfield Strait and around the WAP and this area is mainly characterized by a high sea ice seasonality and drift ice from the Weddell Sea (Collares et al., 2018). Comparisons of individual biomarker concentrations with satellite sea ice data reveal a weak and positive correlation between IPSO$_{25}$ concentrations and winter sea ice concentrations ($r^2$ = 0.5), while no correlation is found between sea ice and pelagic biomarker concentrations ($r^2$ < 0.1 for all relations). Correlations of PIPSO$_{25}$ values with satellite-derived sea ice concentrations (for spring, summer, autumn and winter) contrast earlier observations made for the PIP$_{25}$ index in the Arctic Ocean, where the closest linear relationship is found mainly with the spring sea ice coverage (i.e. the blooming season of sea ice algae; Müller et al., 2011; Xiao et al., 2015).

Esper and Gersonde (2014a) studied the response of diatom species to changes in environmental conditions and their response to the non-linear behaviour of sea ice dynamics (Zwally et al., 2002). In contrast to ice free areas or areas of permanent sea ice cover, areas characterized by the transition from consolidated to unconsolidated sea ice show rapid changes in satellite derived sea ice concentrations (ranging from 90 % to 15 %) and exhibit a large variability in species composition. To reflect this curve in sea ice we hence chose a cubic polynomial regression (polynomial of third degree) to determine the relation between PIPSO$_{25}$ values and satellite data depicting sea ice concentrations of more than 20 %.

We observe a remarkably low correlation between PIPSO$_{25}$ values and spring sea ice concentrations of less than 20 % with a coefficient of determination $r^2$ = 0.37 for P$_Z$IPSO$_{25}$, $r^2$ = 0.50 for P$_E$IPSO$_{25}$ (Fig. 5a), $r^2$ = 0.31 for P$_B$IPSO$_{25}$, and $r^2$ = 0.34 for P$_D$IPSO$_{25}$ (Fig. 5b). The highest correlation is observed between winter sea ice concentrations and P$_E$IPSO$_{25}$ ($r^2$ = 0.72), and P$_Z$IPSO$_{25}$ ($r^2$ = 0.65, Fig. 5c). A weaker correlation is noted for the sterol-based PIPSO$_{25}$ values (P$_B$IPSO$_{25}$: $r^2$ = 0.52; P$_D$IPSO$_{25}$: $r^2$ = 0.44, Fig. 5d). The slightly sigmoid-shaped regression line of winter sea ice concentrations and PIPSO$_{25}$ values reflects the non-linearity of sea ice cover in different sea ice regimes as mentioned above.

[revised manuscript text omitted]

We applied the transfer function of Esper and Gersonde (2014a) with four analogs (4an, Table 4) to our samples to compare the different estimates of sea ice cover based on biomarkers and diatoms. A positive correlation of the linear relationship is found between WSI concentrations derived from diatoms and the PIPSO$_{25}$ indices based on HBI trienes (P$_Z$IPSO$_{25}$ with $r^2 = 0.76$; P$_E$IPSO$_{25}$ with $r^2 = 0.77$, Fig. 6a). The correlations of sterol-based PIPSO$_{25}$ values with WSI are slightly lower but in the same range (P$_B$IPSO$_{25}$ with $r^2 = 0.74$; P$_D$IPSO$_{25}$ with $r^2 = 0.69$, Fig. 6b). A slightly weaker correlation is noted for diatom- and satellite-based winter sea ice concentrations ($r^2 = 0.63$; Fig. 6c). Overall, the diatom approach indicates higher sea ice concentrations than the satellite data with an offset of up to 65 %. This may be due to different sources of satellite reference data used for the transfer function or also due to the fact that the sediment samples integrate a longer time period with a higher sea ice cover than the satellite data (see discussion in section 4.3).

**4.5 Application of PIPSO$_{25}$ as a semi-quantitative sea ice index**

Precise and, in particular, quantitative reconstructions of past sea ice coverage are crucial for a robust assessment of feedback mechanisms in the ice-ocean-atmosphere system. While diatom transfer functions provide a valuable tool, additional information on sea ice conditions in coastal ice-shelf proximal areas, which are often affected by opal dissolution, are essential. The PIPSO$_{25}$ approach seems to be a promising step into this direction, though our data obtained for the WAP are not yet sufficient for a full calibration. PIPSO$_{25}$, diatom and satellite sea ice data, however, reveal positive correlations (Figs. 5 and 6) and depict similar gradients in sea ice cover. The observed offset between satellite data and biomarker- and diatom-based sea ice estimates likely relates to the fact that the instrumental records cover a significantly shorter or more recent time interval than the studied sediments. The recent rapid warming along the WAP (Vaughan et al., 2003) hence complicates attempts to calibrate these proxy data against observational data. Regarding the interpretation of PIPSO$_{25}$ in terms of sea ice coverage in the study area, lower PIPSO$_{25}$ values (<0.15 for P$_Z$IPSO$_{25}$; <0.31 for P$_E$IPSO$_{25}$; <0.22 for P$_B$IPSO$_{25}$ and P$_D$IPSO$_{25}$) roughly seem to reflect unconsolidated, drifting winter sea ice and a nearly ice-free spring season. Higher values (>0.71

for P$_Z$IPSO$_{25}$; >0.9 P$_E$IPSO$_{25}$; >0.6 for P$_B$IPSO$_{25}$ and P$_D$IPSO$_{25}$) would refer to an extended winter sea ice cover (up to 91 % in some years) with ice floes remaining until summer.

**5    Conclusion**

[revised manuscript text omitted]
/080-2 | 35.1 | 0 | 0 | 0 | 0.7 | 0 | 0 | 0 | 0 | 0 | 0 | 0 | 0 | 0 | 0 | 0 | 0 | 0 | 0 | 0 |

---

## Author Response (AR1)

Dear Marcel van der Meer, thank you very much for your constructive comments encouraging revision and restructuring of our manuscript. We considered all your comments carefully and we believe that our revised manuscript now clearly addresses all your requests. Please find below our responses (regular fonts) to your comments (bold fonts).

**First of all I would like to thank both reviewers for reviewing this manuscript and you for your reply. I think the reviewers made some good suggestions concerning your manuscript and I think you should definitely address these.**

- We addressed all questions and comments raised by both reviewers (see response letters and modified manuscripts uploaded to the discussion forum in February and May 2019, respectively) and are confident that our revisions satisfy their requests. While revising our manuscript again to address the editor's most recent recommendations, previous modifications suggested by the reviewers are still valid.

**There are two main issues I think you really need to address. The first issue deals with the analytical details. In response to my earlier comments you have already addressed some. I assume that when silylating sterols you added both BSTFA and pyridine, for instance?**

- We note that the addition of pyridine for sterol silylation is not part of our (and e.g. Simon Belt's) analytical protocol. We revised the methods section (section 3.2) and added references for our laboratory analyses. We note that our sterol treatment and analysis follow the procedure outlined in Belt et al. (2013) and Stein et al. (2012) to ensure comparability of the results obtained within different laboratories.

**Another question I have related to analytical details is if you could give a more thorough description on how you calibrated the HBI response on the MS in order to be able to compare them with other compounds?**

- We revised this in section 3.2 and added information on the determination of instrumental response factors by means of a standard sediment with known (GC-FID determined) HBI concentrations. We now also include details on the HBI identification and quantification in the supplementary material.

**How does this affect the use of the c-factor, sometimes you use it, sometimes you do.**

- Previously, the so-called c-factor has been introduced to account for the normally much higher concentrations of phytosterols compared to HBI concentrations. Considering this factor for the calculation of sterol-based PIP25/PIPSO25 indices clearly facilitates their comparison with HBI-triene-based PIP25/PIPSO25 indices (using a c-factor of 1; Smik et al., 2016). Sterol-based PIP25/PIPSO25 values determined without a c-factor would be orders of magnitude lower. We rephrased the sections commenting on the application of the c-factor within the methods section (3.2) and now provide references on (ongoing) discussions on the application of the c-factor (e.g., Smik et al. (2016), Müller et al. (2011) and Belt and Müller (2013)).

**There is a relatively large number of "zero" points, how does this affect your correlations?**

- Most of these "zero points" relate to sampling stations in the Drake Passage, where the absence of IPSO25 (and PIPSO25 values of zero) are in accordance with the absence of sea ice. We comment on the absence of IPSO25 at two core sites off the continental slope which (according to satellite data) experience sea ice cover in section 4.2. We relate the absence of IPSO25 at these sites to e.g. the environmental preferences of its source diatom *Berkeleya adeliensis* (restricted to landfast/platelet ice). Accordingly, also PIPSO25 values are zero at these sites (suggesting ice-free conditions). For the correlation of biomarker data (+ PIPSO25 values) with satellite data, we follow earlier studies (Müller et al., 2011; Navarro-Rodriguez et al., 2012; Smik et al., 2016; Xiao et al., 2015) and argue that "zero points" referring to the absence of sea ice should not be excluded (in our study, omitting these "zero points" would lead to weaker correlations; see PIPSO25 correlations with spring and winter sea ice below).

[Figure]

**How do you deal with the large variability or standard deviation in your sea ice concentration?**

- We now comment on the high standard deviation in the satellite-derived sea ice concentrations in the method section 3.4 and consider that the use of mean averages facilitates the comparison with sedimentary data (integrating and reflecting a significantly longer time interval of variable sea ice conditions).

**My second issue has to do with the set-up of the manuscript. You now discuss the data in light of your new sea ice proxy PIPSO25, the alternative would be to discuss your data in light of all the environmental parameters including sea ice and end with by suggesting a potential new sea ice proxy for the Antarctic region, PIPSO25. Basically reorganizing the discussion to a more general discussion rather than the focus on the proxy. The data you have generated is clearly interesting, the issues are with the PIPSO proxy, by shifting the focus you would generate a more acceptable manuscript. It might be that the oceanographic setting around Antarctica is so complex that a sea ice proxy similar to the Arctic is just not feasible, for instance. When discussing your data in light of the oceanography, sea ice extend and other environmental parameters this outcome would be fine. In the current manuscript with the focus on the PIPSO25 proxy, the proxy has to work, any other conclusion cannot not be accepted. I think this is why the manuscript as is feels a bit forced, PIPSO25 has to work and this is not necessary.**

- We agree with the editor and restructured the results/discussion chapters as recommended. First, we now present and discuss the distribution of biomarkers in the light of sea surface conditions (i.e. oceanographic features; primary productivity; (potential) source organisms for the individual lipids). These sections are now followed by a comparison of the biomarker distribution patterns with satellite sea ice data and diatom-based sea ice estimates. In the final results/discussion chapter (4.4) we now present and discuss the adaptation of the Arctic Ocean PIP25-approach using IPSO25 for calculating the PIPSO25 index. We now also comment on the role that platelet ice formation plays for the distribution of IPSO25 and the potential application of PIPSO25. Clearly, the sea ice environment in the Southern Ocean differs from the Arctic Ocean and these limitations have to be considered for an attempt to adopt the semi-quantitative PIP25 approach. We now emphasize these limitations in the manuscript.

**Please also check if you site all relevant literature, also the more recent publications and in the right place.**
- We revised and updated the literature data base and now also include most recently published papers.

**Again, with a slightly different focus I think you have a great dataset and a very interesting manuscript. I am looking forward to your rebuttal and revised manuscript.**
- We are very glad that you appreciate our work and hope that our recent revisions fulfill your expectations.

IPSO$_{25}$ concentrations more insight into the production and sedimentation of the involved biomarker lipids is needed to develop such a semi-quantitative approach.

With regard to the spatially and temporally variable sea ice extent, Esper and Gersonde (2014a) studied the response of diatom species to changes in environmental conditions and their response to the non-linear behavior of sea ice dynamics (Zwally et al., 2002). In contrast to ice free areas or areas of permanent sea ice cover, areas characterized by the transition from consolidated to unconsolidated sea ice show rapid changes in satellite derived sea ice concentrations (ranging from 90 % to 15 %) and exhibit a large variability in species composition. To reflect this curve in sea ice we hence chose a cubic polynomial regression (polynomial of third degree) to determine the relation between PIPSO$_{25}$ values and satellite data depicting sea ice concentrations of more than 20 %. A

slightly sigmoid-shaped regression line of winter sea ice concentrations and PIPSO$_{25}$ values depicts the non- linearity of sea ice cover in different sea ice regimes.

A positive correlation is found between WSI concentrations derived from diatoms and the PIPSO$_{25}$ indices based on HBI trienes ($P_Z$IPSO$_{25}$ with $r^2 = 0.76$; $P_E$IPSO$_{25}$ with $r^2 = 0.77$, Fig. 8a). The correlations of sterol-based PIPSO$_{25}$

values with WSI are slightly lower but in the same range ($P_B$IPSO$_{25}$ with $r^2 = 0.74$; $P_D$IPSO$_{25}$ with $r^2 = 0.69$, Fig.

[revised manuscript text omitted]

**Application of PIPSO$_{25}$ as a semi-quantitative sea ice index**¶
Precise and, in particular, quantitative reconstructions of past sea ice coverage are crucial for a robust assessment of feedback mechanisms in the ice-ocean-atmosphere system. While diatom transfer functions provide a valuable tool, additional information on sea ice conditions in coastal ice-shelf proximal areas, which are often affected by opal dissolution, are essential. The PIPSO$_{25}$ approach seems to be a promising step into this direction, though our data obtained for the WAP are not yet sufficient for a full calibration. PIPSO$_{25}$, diatom and satellite sea ice data, however, reveal positive correlations (Figs. 5 and 6) and depict similar gradients in sea ice cover. The observed offset between satellite data and biomarker- and diatom-based sea ice estimates likely relates to the fact that the instrumental records cover a significantly shorter or more recent time interval than the studied sediments. The recent rapid warming along the WAP (Vaughan et al., 2003) hence complicates attempts to calibrate these proxy data against observational data. Regarding the interpretation of PIPSO$_{25}$ in terms of sea ice coverage in the study area, lower PIPSO$_{25}$ values (<0.15 for P$_Z$IPSO$_{25}$; <0.31 for P$_E$IPSO$_{25}$; <0.22 for P$_B$IPSO$_{25}$ and P$_D$IPSO$_{25}$) roughly seem to reflect unconsolidated, drifting winter sea ice and a nearly ice-free spring season. Higher values (>0.71 for P$_Z$IPSO$_{25}$; >0.9 P$_E$IPSO$_{25}$; >0.6 for P$_B$IPSO$_{25}$ and P$_D$IPSO$_{25}$) would refer to an extended winter sea ice cover (up to 91 % in some years) with ice floes remaining until summer.
------------------------------Seitenumbruch------------------------------
¶
**Conclusion**¶

[revised manuscript text omitted]
/080-2 | 35.1 | 0 | 0 | 0 | 0.7 | 0 | 0 | 0 | 0 | 0 | 0 | 0 | 0 | 0 | 0 | 0 | 0 | 0 | 0 | 0 |

Supplementary Material

[Figure]

**Supplement S1: Examples of mass spectra of IPSO$_{25}$ (m/z 348), HBI Z-triene and E-triene (both m/z 346)**

**obtained from surface sediments in the study area.**

[Figure]

**Supplement S2: Example calibration curve for the quantification of $IPSO_{25}$. Different (true) $IPSO_{25}$**

**concentrations determined via gas chromatography-flame ionization are plotted against (raw) $IPSO_{25}$**

**concentrations determined via gas chromatography-mass spectrometry using selected ion monitoring (m/z**

**348). The instrumental response factor is obtained from the regression line.**

---

## Author Response (AR2)

Dear Marcel van der Meer,

Thank you for your positive response to our revised manuscript. We considered all your comments on minor revisions and changed/revised these parts. We also proof read the manuscript to eliminate typos and misleading phrases.

*I had one or two minor things. At the end of the abstract, is this specific for this area of the world? Perhaps add "in this area" all the way at the end.*
➢ We changed this sentence.

*Page 6 line 28, I would use increase rather than rise.*
➢ We changed the wording accordingly.

*Page 7 line 2, things are usually absent from and not in, if they were in they wouldn't be absent.*
➢ We corrected this.

*Page 7 line 11, total organic carbon content, lose the s.*
➢ We removed the s.

*Page 7 line 26, technically it's a monitoring gas, the external standard is your reference.*
➢ We corrected this.

*Page 11 line 8/9, conditions "for their producers" () "for their source diatoms". One or the other, not both.*
➢ We changed the wording.

*Page 17 line 19-29, this first sentence is a bit weird, normally you would start with the second sentence and than refer to the figure. Now it feels like a paragraph in the discussion starts with part of a figure legend? Personally I would try to avoid that, not a major issue.*
➢ We accordingly changed this paragraph.

*I would go through the text carefully again, make sure there are no typo's or other weird things, but I think this was it.*
➢ We read the manuscript again and corrected typos.